# Bidirectional yet asymmetric causality between urban systems and traffic dynamics in 30 cities worldwide

Yatao Zhang [1,2,3] ✉, Ye Hong [2], Song Gao [4] & Martin Raubal [1,2]

Understanding how urban systems and traffic dynamics co-evolve is crucial for advancing sustainable and resilient cities. However, their bidirectional causal relationships remain underexplored due to challenges of simultaneously inferring spatial heterogeneity, temporal variation, and feedback mechanisms. Here we present a spatio-temporal causality framework that bridges correlation and causation by integrating spatio-temporal weighted regression with spatio-temporal convergent cross-mapping. Characterizing cities through urban structure, form, and function, the framework uncovers bidirectional causal patterns between urban systems and traffic dynamics across 30 cities on six continents. Our findings reveal asymmetric bidirectional causality, with urban systems exerting stronger influences on traffic dynamics than the reverse in most cities. Urban form and function shape mobility more profoundly than structure, even though structure often exhibits higher correlations. This does not preclude the reversed causal direction, whereby long-established mobility patterns can also reshape the built environment over time. Finally, we identify three causal archetypes: tightly coupled, pattern-heterogeneous, and workday-attenuated, which support city-to-city learning and inform context-sensitive strategies in sustainable urban and transport planning.

Cities function as complex systems where human mobility and the built environment continuously interact[1–4]. These interactions manifest in diverse ways within urban traffic systems. For example, transit-integrated urban design in Singapore facilitates individual movement through extensive public transport networks, whereas London's historic and fragmented road layout amplifies congestion-prone traffic patterns. Accordingly, traffic dynamics serve as a tangible indicator of these interactions, reflecting not only how urban systems shape movement patterns but also how human mobility reciprocally influences the built environment[5–7]. Understanding these bidirectional influences is crucial for designing adaptive transportation policies and promoting sustainable urban development.

Given the complexity and multifaceted nature of cities, comprehensively characterizing urban systems is an important prerequisite for investigating these bidirectional influences[4,8]. Here, we characterize cities through three interconnected components: urban structure, form, and function, collectively forming the "triple helix" of urban systems. Urban structure establishes the physical backbone of a city, primarily through its road networks[9,10]. Building on this structural foundation, urban form determines the spatial layout and configuration of the city, while urban function embeds the distribution of land uses and socioeconomic activities across these spatial arrangements[11–13]. Although closely related, they capture different facets of urban systems that influence traffic dynamics through

[1]Future Resilient Systems, Singapore-ETH Centre, ETH Zurich, Singapore, Singapore. [2]Institute of Cartography and Geoinformation, ETH Zurich, Zurich, Switzerland. [3]Department of Geography, University College London, London, UK. [4]Geospatial Data Science Lab, Department of Geography, University of Wisconsin-Madison, Madison, WI, USA. ✉e-mail: yatzhang@ethz.ch; yatao.zhang@ucl.ac.uk

distinct mechanisms[3,14,15]. For example, road network topology constrains route choice, whereas district form and function drive the demand for travel orientations and destinations. Together, these three components govern how urban systems interact with human mobility, shaping the spatial and temporal patterns of traffic dynamics.

Existing studies have primarily examined associations among these components, applying such insights to develop analytical models for urban dynamics, traffic forecasting, and public transit systems[1,6,16,17]. In particular, built-environment factors, ranging from urban morphology indices and land-use types to network structure, have been shown to exert significant influences on traffic states and congestion patterns across diverse metropolitan contexts[15,18–20]. To investigate these complex interactions between urban systems and traffic dynamics, many methodological approaches have been adopted, including global statistical and econometric models for uncovering macro-level determinants and longitudinal dependencies[15,19], local and spatio-temporal varying-coefficient models for characterizing spatiotemporal heterogeneity and dynamic associations[21,22], and computational data-driven frameworks for capturing nonlinear dynamics and fine-grained mobility patterns[2,3,23]. Furthermore, such methodological advances have been deployed in detailed case studies to quantify route diversification capabilities within network topologies[24], assess the influence of road functions on localized congestion[25], and evaluate the impacts of street design and intersection density on traffic performance[18]. Collectively, these studies establish important empirical benchmarks on the interplay between urban systems and traffic dynamics, underscoring that both supply-side factors (e.g., network structure) and demand-side factors (e.g., land-use and activity patterns) within the built environment are critical determinants of traffic states. However, the bidirectional feedback that governs these interactions and the potential asymmetries in their strength remain underexplored, despite being central to their dynamic and reciprocal nature. For example, while efficient road planning can mitigate traffic congestion, persistent congestion can also reshape urban systems, as seen in cities that expand metro networks in response to traffic pressure[26]. Without accounting for this feedback, developed models may lead to an underestimation of long-term shifts in mobility and infrastructure demands.

Understanding these feedback mechanisms requires moving beyond statistical associations toward causal modeling that addresses both spatial dependency and temporal variation[27,28]. Causal inference has proven effective in capturing interactions among diverse elements of urban and traffic systems, ranging from linking individual mobility to health outcomes[29], to diagnosing congestion in road networks[30], and disentangling complex urban processes[31]. Among these, convergent cross mapping (CCM) has emerged as a robust tool for detecting nonlinear and bidirectional causal relationships in urban contexts[30,32]. However, most existing studies focus on investigating causal inference in time-series data to establish temporal causality within the built environment. In contrast, spatial causality modeling remains relatively underexplored, largely due to challenges in modeling spatial dependencies that underlie geographical processes[33]. A feasible solution is to adapt causal inference models to the spatial domain by integrating spatial dependencies in urban space, as exemplified by the geographical convergent cross mapping (GCCM) framework[34]. However, GCCM does not explicitly account for temporal changes that are fundamental to urban systems. As features characterizing urban structure, form, and function evolve gradually, movement patterns fluctuate much more rapidly, producing pronounced temporal variations in urban dynamics[3]. In addition, causal mechanisms are inherently spatially heterogeneous, varying substantially across cities[4,35,36]. This spatial adaptability of causality complicates the bidirectional interactions between urban systems and traffic dynamics, which highlights the need for an integrated spatio-temporal framework to enhance context-aware understanding of

urban dynamics and foster city-to-city learning networks for knowledge exchange.

To address these challenges, we propose a spatio-temporal causality framework that bridges correlation and causation by identifying bidirectional causal relationships between urban systems and traffic dynamics. At its core is a spatio-temporal convergent cross-mapping method (STCCM) developed in this study, which extends CCM and GCCM by jointly embedding spatial dependence and temporal variation into the state-space reconstruction. Intuitively, STCCM uncovers causal relationships by examining whether the spatio-temporal evolution of one process, such as traffic dynamics during congestion periods, embeds information that allows the recovery of another process within the joint state space of interacting urban systems. The framework combines STCCM with spatio-temporal weighted regression (STWR) (see "Methods"). In detail, STWR first characterizes the spatially heterogeneous and temporally varying relationships between urban systems and traffic dynamics, and then synthesizes these locally estimated effects into composite indicators that capture the combined influence of urban structure, form, and function. Subsequently, STCCM uses these composite indicators to infer nonlinear and bidirectional causal pathways during congestion periods, when mobility-infrastructure mismatches amplify causal signals and make them more detectable and policy-relevant. This integrated pipeline generates comparable causal fingerprints across space and time, converting heterogeneous associations into directional evidence of causality. To ensure broad representativeness, we apply this framework to 30 cities with diverse urban contexts, selected based on data availability, traffic congestion levels, and global distribution (see "Methods"). These cities encompass major metropolitan capitals in developed countries and economic centers in emerging economies, spanning six continents. Finally, we introduce a causality-driven clustering approach that integrates all three urban system components to reveal underlying similarities and differences in bidirectional causal patterns across cities (see "Methods"). This analysis identifies distinct causal archetypes and supports city-to-city learning, outlining scalable pathways from causal insights to intervention for advancing sustainable urban and transport planning.

## Results

### Associating urban systems with traffic dynamics

The three components of urban systems exert distinct influences on traffic dynamics, with varying effects across spatial and temporal dimensions. To assess these effects, we characterize urban systems using distinct yet complementary interpretable feature sets, with urban structure depicted by road network topologies, urban form captured through morphological metrics at the landscape level, and urban function represented by land-use attributes (see "Methods"). Traffic dynamics are measured using a congestion indicator that captures variations in human mobility across space and time within the road network (see "Methods"). The STWR model is then employed to quantify these relationships, allowing the strength of association to vary across space and time rather than imposing a single global effect. In Fig. 1a, we present the $R^2$ values across 30 cities on rest days for urban structure, form, and function, encompassing their average values, as well as ± 1 standard deviation. Higher $R^2$ values indicate a better fit of the STWR model. See Supplementary Tables 1 and 3 for model fit ($R^2$), local significance assessment (adjusted $\alpha$ and critical $t$), and model selection (Akaike Information Criterion corrected, AICc) across 30 cities on rest days. Overall, all three components exhibit strong and significant correlations with traffic dynamics, underscoring their integral roles in shaping individual movement patterns. Among them, urban structure demonstrates the strongest association with traffic dynamics, with an average $R^2$ of 0.94 and the first (Q1) and third (Q3) quartiles of 0.92 and 0.97, respectively. This outcome is anticipated, as the urban road network serves as the physical framework-or

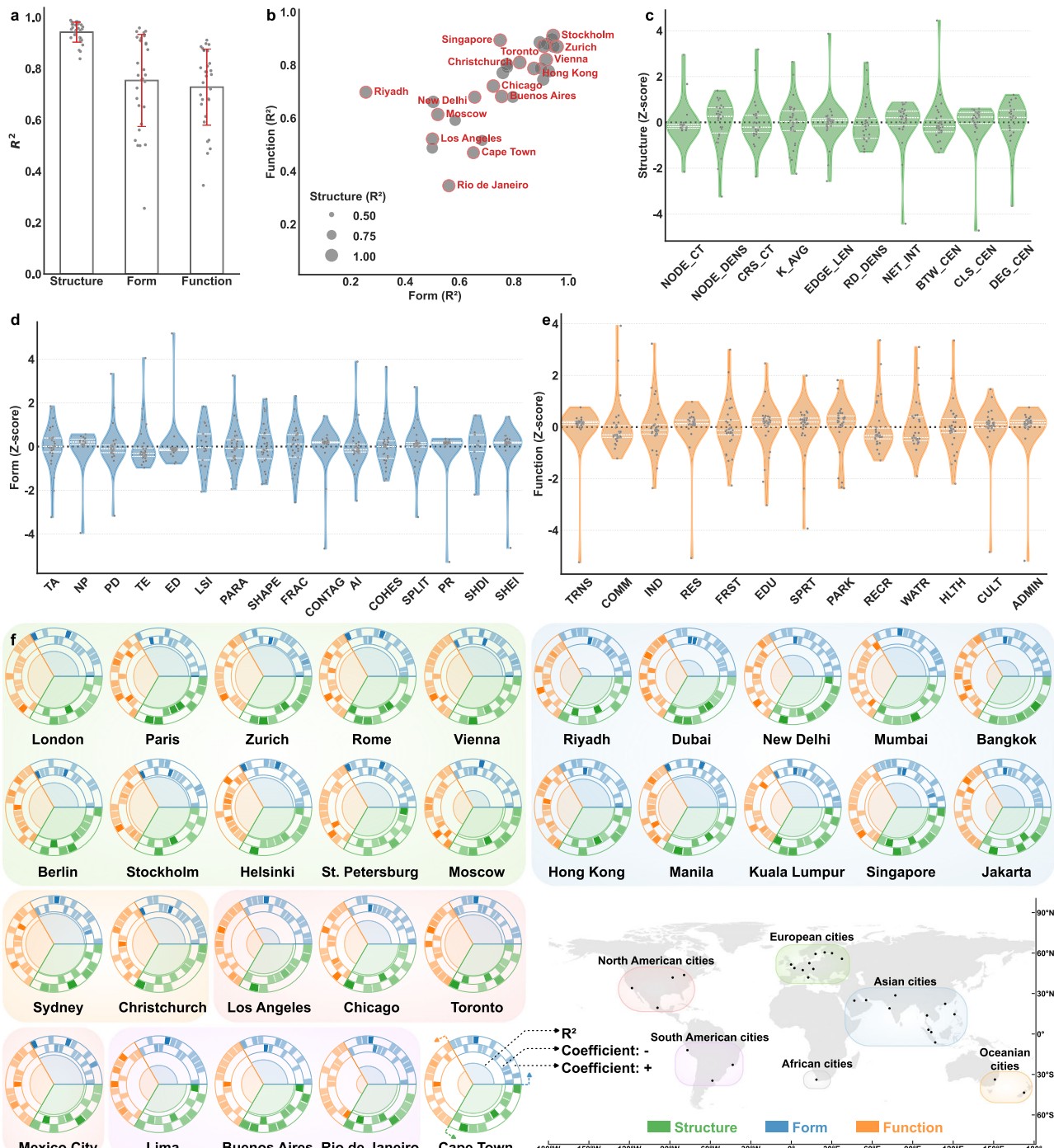

**Fig. 1 | Spatio-temporal correlation between urban systems and traffic dynamics during rest days across 30 cities ($n = 30$ cities). a** Average $R^2$ values across 30 cities, representing the spatio-temporal correlation estimated by the STWR model. Bars indicate mean $R^2$, with error bars representing $\pm 1$ standard deviation (SD). Dots denote individual city-level observations. **b** Bubble plot of city-specific $R^2$ values, where the $x$- and $y$-axes indicate urban form and function, respectively, and the bubble size reflects urban structure. **c–e** Violin plots showing the distribution of STWR coefficients for features derived from urban structure, form, and function, respectively. Definitions of all feature abbreviations are provided in Supplementary Table 9. Inner white lines indicate the 25th, 50th, and 75th percentiles. Dots denote individual city-level observations. **f** Global overview of STWR results across 30 cities, with urban structure, form, and function shown for each city. Cities are visualized as concentric circles: the innermost circle represents the $R^2$ value, the middle ring shows negative STWR coefficients, and the outer ring depicts positive STWR coefficients. Within each feature group, individual features are arranged counterclockwise in the same order as in the violin plots. For example, in urban form, TA (total area) is positioned at the rightmost segment, followed counterclockwise by NP (number of patches) and ending with SHEI (Shannon's evenness index, the evenness of area distribution among patch types). Administrative boundaries are derived from the Global Administrative Boundaries (GADM).

"container"-of traffic, connecting various parts of the city and facilitating movement[10]. Its foundational role as the spatial backbone of urban systems accounts for this high degree of correlation. Urban form and function display similar levels of association with traffic dynamics, albeit slightly lower than urban structure. Urban form attains an average $R^2$ of 0.75, closely followed by urban function at 0.73. While urban structure establishes the spatial framework of the city, urban form and function populate this container, providing the spatial

layouts and land use configurations that drive human mobility[14,37]. Together, they influence the traffic system by shaping the patterns and intensity of movement across the city.

Despite the overall significance, the three urban system components display considerable variation in their $R^2$ values across cities, as shown in Fig. 1b, f. These disparities appear to be partly shaped by underlying socioeconomic conditions and governance context[38], and also by the extent to which the globally consistent feature set captures locally salient determinants of traffic patterns. Several high-income cities demonstrate high $R^2$ values across all three components, such as Stockholm, Zurich, and Toronto. Notably, they also exhibit the highest $R^2$ values among the 30 cities for individual components: Berlin records the highest value for urban structure at 0.99, Zurich achieves the highest $R^2$ value for urban form at 0.96, and Stockholm leads in urban function at 0.91. These high $R^2$ values suggest a tighter coupling between urban systems and traffic dynamics, particularly in cities with advanced infrastructure and stable institutional conditions. In contrast, cities in less-developed countries, such as Rio de Janeiro, Riyadh, and Cape Town, exhibit much lower $R^2$ values for all three components. For example, the $R^2$ value between urban form and traffic dynamics in Riyadh is merely 0.26, while Rio de Janeiro shows a low $R^2$ of 0.35 for urban function. In these contexts, the lower $R^2$ indicates that the form and function features used here account for a smaller share of the variance in traffic patterns. This may reflect weaker associations with spatial configurations and land-use distributions, or the influence of latent factors not captured by the current feature set used to construct the composite indicator. However, $R^2$ values do not entirely correspond to local socioeconomic development, as evidenced by exceptions like Manila and Los Angeles. Manila's compact and road-based mobility context tightens the alignment of urban systems with traffic dynamics, whereas Los Angeles's polycentric sprawl and extensive boundaries weaken form- and function-traffic links, leaving structure as the dominant correlate.

The variations in $R^2$ values among different cities can be further explained through the STWR coefficient distribution in Fig. 1c–e. These coefficients highlight regional heterogeneity in the interactions between urban system features and traffic dynamics, measured by traffic congestion levels. Some features exhibit both positive and negative effects across cities, whereas others consistently influence movement patterns in the same direction. Urban structure features in Fig. 1c generally present mixed effects on traffic dynamics, which reflects the complex interplay between road network design, topology, and regional characteristics. For example, metrics such as CRS_CT (crossing count, the number of road intersections) and RD_DENS (road density, the total road length per unit area) may have positive or negative impacts depending on whether they enhance connectivity or create bottlenecks in specific regions. For urban form features in Fig. 1d, NP (number of patches, the total number of land-use patches) and CONTAG (contagion, the degree to which land-use patches are clumped or dispersed) positively correlate with traffic congestion due to increased fragmentation and uneven spatial aggregation, which could hinder connectivity and intensify traffic density. In contrast, TE (total edge, the total length of all patch edges) and ED (edge density, the total length of all patch edges per unit area) generally show negative effects, since more extensive edge configurations can improve spatial connectivity. Most of the other features exhibit mixed effects, with their influence varying across cities. For urban function features in Fig. 1e, categories such as TRANS (transportation area), SPRT (sports area), and PARK (park area) display predominantly positive effects on traffic dynamics. These land uses are associated with destinations that attract leisure-related mobility on rest days. In contrast, COMM (commercial and business facilities area) presents an overall negative effect, potentially reflecting reduced activity that leads to decreased traffic flows during non-working periods.

When features are considered collectively, their interactions and localized weighting can shift the coefficients between positive and negative values. Within a single city, spatial heterogeneity can also lead to localized variations in effects. Corresponding STWR results for workdays are provided in Supplementary Fig. 1 and Supplementary Tables 2 and 4.

## Revealing bidirectional causal patterns

Recognizing that correlation does not imply causation, it is crucial to move beyond statistical associations and identify causal pathways[27,39]. To capture the bidirectional causal relationships between urban systems and traffic dynamics, we develop the spatio-temporal convergent cross mapping (STCCM) model by incorporating spatial dependency and temporal variation (see "Methods"). The core principle is to determine whether spatially distributed information that evolves over time in one variable can predict another by reconstructing their state spaces, thereby inferring causality. This bidirectional cross-mapping approach reveals the directional influence and relative strengths. Figure 2 presents the inferred bidirectional causal patterns between urban systems and traffic dynamics across 30 cities during congested periods on rest days, with detailed performance metrics and significance tests reported in Supplementary Table 5. Each $L$-$\rho$ plot serves as a diagnostic tool for causal inference, illustrating how predictability evolves with increasing library size $L$. Here, $L$ represents the number of observations used for state-space reconstruction, while $\rho$ represents the cross-mapping skill of one variable based on another, i.e., how well one variable recovers the state of the other. For each city and each component of the urban system, we generate two $L$-$\rho$ curves: one for urban systems → traffic dynamics and one for traffic dynamics → urban systems. Crucially, evidence of causality is established not by a static $\rho$ value, but by the property of convergence: a curve that rises and stabilizes at higher $\rho$ values as $L$ grows. This trend indicates a robust causal signal, as additional observations improve coverage of the reconstructed state space and help reveal the underlying deterministic rules of the system[34]. Furthermore, the shaded regions between the two curves in Fig. 2a highlight the magnitude of asymmetry ($\Delta\rho$) between the bidirectional influences. Consequently, for a given urban system indicator $X$ and traffic dynamics indicator $Y$, if the $\rho$ values for $X \rightarrow Y$ consistently exceed those for $Y \rightarrow X$ as $L$ increases, $X$ is inferred to be the dominant causal driver of $Y$, and vice versa.

Figure 2b–d highlights a pronounced bidirectional yet asymmetric causal pattern, exemplified by Singapore, where $\rho$ values reach exceptionally high levels at the largest library size. With increasing $L$, the $L$-$\rho$ curves for urban structure, form, and function → traffic dynamics consistently exceed those for the reversed direction, i.e., traffic dynamics → urban structure, form, and function. This indicates a dominant causation from urban systems to traffic dynamics, with spatial layouts and road networks shaping traffic congestion patterns. However, this does not negate the existence of reversed causation. The relatively slower, yet still high, $\rho$ values for traffic dynamics → urban systems reflect feedback processes between mobility and the built environment[5,40]. In Singapore, urban planning and governance significantly shape human mobility patterns, while these patterns in turn influence urban development through dynamic responses, such as promoting public transport and integrating land-transport planning to mitigate road congestion[41]. In addition, among the three urban system components, urban form exerts the strongest influence on traffic dynamics. Its $L$-$\rho$ curve reaches the highest $\rho$ values, highlighting its key role in shaping traffic patterns and driving congestion. This finding diverges from the STWR model results in Fig. 1f, where urban structure demonstrated the highest $R^2$ value in Singapore. This divergence underscores the critical distinction between association and causation, emphasizing that strong correlations do not necessarily imply causal relationships.

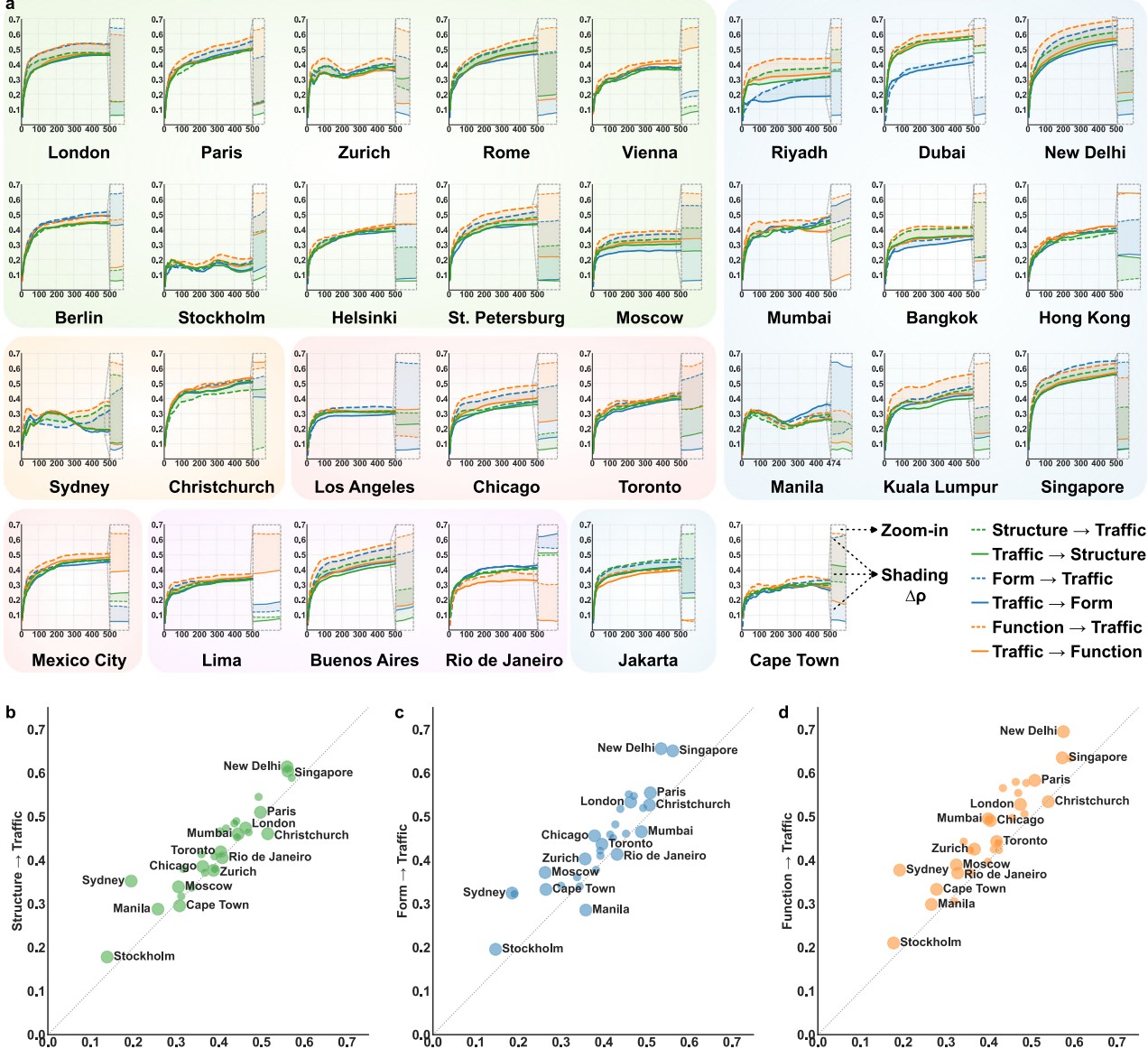

**Fig. 2 | Bidirectional causal patterns between urban systems and traffic dynamics during congestion periods on rest days. a** $L$-$\rho$ plots from the STCCM model for each city, represented by urban structure, form, and function. Increasing $\rho$ values with larger library sizes $L$ indicate the presence of spatial causality. The enlarged view on the right magnifies the trends at large library sizes to clearly reveal the directional difference in causal strength, while the shaded regions between the curves highlight the magnitude of asymmetry ($\Delta\rho$) between the bidirectional causal influences. Average $\rho$ values at the largest $L$ for all 30 cities depict bidirectional causal relationships between traffic dynamics and **b** urban structure, **c** form, and **d** function. Each point represents a city, with the $x$-axis indicating the $\rho$ value for traffic dynamics → urban systems and the $y$-axis indicating the $\rho$ value for urban systems → traffic dynamics. The dotted 45-degree line marks directional differences: cities above the line exhibit stronger causality from urban systems to traffic dynamics, while those below indicate stronger causality in the reverse direction. Note that all plotted cities exhibit statistically significant causal relationships ($\rho$) at the $p < 0.001$ level. Detailed significance tests for both $\rho$ and the directional asymmetry ($\Delta\rho$) are provided in Supplementary Tables 5 and 7.

Similar to Singapore, several other cities exhibit comparable bidirectional yet asymmetric causal patterns between urban systems and traffic dynamics during congestion periods. For example, New Delhi, Paris, and London are positioned near Singapore in Fig. 2b–d, with a stronger causal direction of urban systems → traffic dynamics compared to the reversed direction. Their high $\rho$ values at the largest $L$ reveal that their spatial configurations and structures play a significant role in driving traffic congestion. Meanwhile, cities such as Stockholm, Sydney, and Moscow, positioned toward the left in Fig. 2b–d, also indicate a stronger causal direction from urban systems to traffic dynamics. However, their $\rho$ values are considerably lower than Singapore's. This suggests that while urban systems still influence traffic

congestion in these cities, the strength of the causal pathway in both directions is relatively weak. The variations in $\rho$ values across cities can be attributed to city-specific urban configurations and how effectively the STCCM model reconstructs their spatial and temporal interactions in the state space. For cities with higher $\rho$ values, the causal model produces more representative state-space reconstructions, effectively incorporating spatially lagged information across the built environment to capture causal interactions between urban systems and traffic dynamics. In contrast, cities with lower $\rho$ values demonstrate less representative state-space reconstructions, influenced by weaker spatial dependencies and increased variability in the built environment and individual movement patterns. Although some cities near the

diagonal might suggest a balanced interaction, statistical tests support the robustness of these relationships. Specifically, the estimated causal strengths ($\rho$) are statistically significant across all 30 cities ($p < 0.001$; see Supplementary Table 5). Furthermore, despite the visual proximity to the diagonal, paired t-tests reveal that the asymmetry ($\Delta\rho$) remains statistically significant for the majority of cities on rest days, specifically in 26, 29, and 27 out of 30 cities for urban structure, form, and function, respectively (see Supplementary Table 7). Together, these results statistically support the conclusion that urban systems more often exert a dominant influence on traffic dynamics, consistent with the asymmetric nature of the causality.

In addition, rare cases such as Manila and Christchurch reveal distinct bidirectional causal patterns, in which the dominant causal direction runs from traffic dynamics to urban systems. With relatively low $\rho$ values of approximately 0.3, Manila appears below the diagonal line in Fig. 2c but lies above the one in Fig. 2b, d. This pattern highlights its stronger causality directions of urban form $\rightarrow$ traffic dynamics and traffic dynamics $\rightarrow$ urban structure/function, as opposed to the reversed directions. Its $L$-$\rho$ curves show that the relationships are not monotonically increasing with library size (see Fig. 2a). Instead, the curves exhibit slight dips before rising, reflecting weak spatial dependencies in capturing the causal-effect relationship between urban systems and traffic dynamics. In contrast, Christchurch consistently displays monotonic $L$-$\rho$ curves with relatively higher overall $\rho$ values. At the largest library size $L$, the direction of traffic dynamics $\rightarrow$ urban function becomes predominant over its reverse, highlighting the significant feedback effect of mobility patterns on urban land uses. This suggests that human mobility patterns in Christchurch influence and shape how urban functions are utilized and experienced within the city.

Overall, the results underscore the bidirectional causal relationship between urban systems and traffic dynamics, yet reveal a clear asymmetry: urban systems exert stronger and more systematic causal effects on traffic dynamics, whereas feedback from traffic dynamics to urban systems is comparatively weaker. This trend is particularly pronounced on rest days, underscoring the role of urban structure, form, and function in shaping travel behavior. The asymmetry persists on workdays (See Supplementary Fig. 2 and Supplementary Tables 6 and 7), but a slight distinction emerges: stronger traffic-to-urban effects occur more frequently, especially for urban structure. This phenomenon appears in cities such as Manila, Christchurch, Sydney, and Hong Kong, where regular commuting rhythms and higher workday volumes provide discernible feedback to urban systems. It also indicates that long-established mobility patterns can reshape land-use patterns and trigger adjustments in the design and capacity of transport infrastructure.

## Understanding global cities through bidirectional causality

Urban structure, form, and function, together constituting the "triple helix" of urban systems, provide a cohesive lens for linking the built environment to traffic patterns. Categorizing cities based on these interactions can uncover their commonalities and distinctions, supporting city-to-city learning and cross-regional knowledge exchange. To achieve this, we characterize global cities using the shape and magnitude of their $L$-$\rho$ bidirectional causality curves across both rest and work days. Specifically, dynamic time warping (DTW) evaluates the shape difference between the bidirectional $L$-$\rho$ curves of two cities, capturing variations in the temporal patterns of causality. In contrast, the Jaccard distance measures dissimilarity via the intersection-over-union (IoU) of the regions under the bidirectional $L$-$\rho$ curves (defined as the union of the two directional regions), reflecting differences in the overall magnitude of their causal influence. We then integrate these two measures through hierarchical clustering, grouping the 30 cities into three clusters at a distance threshold of 1.0 (see "Methods";

Fig. 3a). To further illustrate inter-city variation, Fig. 3b presents the $\rho$ values at the largest library size $L$ for each city on rest and work days.

Cluster 1 exhibits consistently high $\rho$ values in both causal directions, indicating strong bidirectional causality between urban systems and traffic dynamics. This cluster spans cities from New Delhi to Paris and includes identifiable subgroups such as developing megacities (New Delhi, Mexico City, Mumbai) and global financial hubs (Singapore, Dubai). In these contexts, urban-traffic systems are tightly coupled: landscape morphology, land-use allocation, and road-network topology jointly exert strong influences on traffic dynamics. These robust linkages highlight the strategic value of this cluster for congestion mitigation and sustainable transport planning, as urban system variables already explain a substantial share of spatial and temporal variations in traffic dynamics. Accordingly, practical strategies should prioritize integration of land use and transport planning, coordination across modes and districts through fine-grained street design, and bundling of instruments such as transit provision, pricing schemes, and parking policies[41–43].

Cluster 2 is characterized by relatively strong bidirectional causality, albeit slightly weaker than Cluster 1. This pattern is evident in the $\rho$ values at the largest $L$ in Fig. 3b, spanning cities from Helsinki to Vienna. Within this group, causal patterns are heterogeneous, with several city pairs and triads displaying closely aligned yet distinct causality signatures. For example, Helsinki and Hong Kong show balanced causality magnitudes in both directions for work and rest days; Riyadh and Bangkok are dominated by function- and structure-led patterns, with strong land-use and road-network signals; while Chicago, Toronto, and Vienna share similar rest-day $\rho$ values across components, with consistently lower influence in the reverse direction. This heterogeneity calls for a diagnostic portfolio in planning practice, such as applying the six-direction heatmap to identify component-dominant subareas, including commute corridors shaped by structure and districts influenced by form and function, and then assembling context-specific combinations of network operations, land-use adjustments, and demand management strategies to alleviate congestion[43,44].

Comprising only five cities, Cluster 3 presents uniformly low $\rho$ magnitudes compared to the other clusters. A pronounced workday drop is observed in most cases, except for Stockholm, which shows consistently low values across both rest and work days. Overall, rest-day values remain modest, with cities such as Zurich and Sydney exhibiting higher urban system $\rightarrow$ traffic dynamics than the reverse, while work-day values become consistently lighter in the heatmap, reflecting weaker cross-mapping skill at the largest $L$. This workday attenuation suggests that policies relying solely on static urban system levers are unlikely to yield substantial congestion relief on work days unless complemented by operational and demand-side instruments that directly shape daily dynamics[45,46]. In contrast, rest-day patterns remain more responsive to urban system features, allowing system-based interventions to exert greater influence.

The tendency for higher $\rho$ values on rest days than work days is not limited to Cluster 3, but is broadly observed across most cities, albeit to varying degrees (see Fig. 3b). This can be attributed to two potential factors. First, traffic on rest days is largely self-organized, allowing individuals greater flexibility to adjust their travel behavior in response to congestion. This responsiveness reinforces the role of urban systems in shaping movement choices. Second, during peak workday hours, traffic dynamics tend to approach saturation due to fixed commuting schedules, resulting in congestion across various urban regions. This congestion extends over diverse road networks and land-use types, reducing spatial heterogeneity in causality patterns and rendering causal relationships less distinct than those observed on rest days.

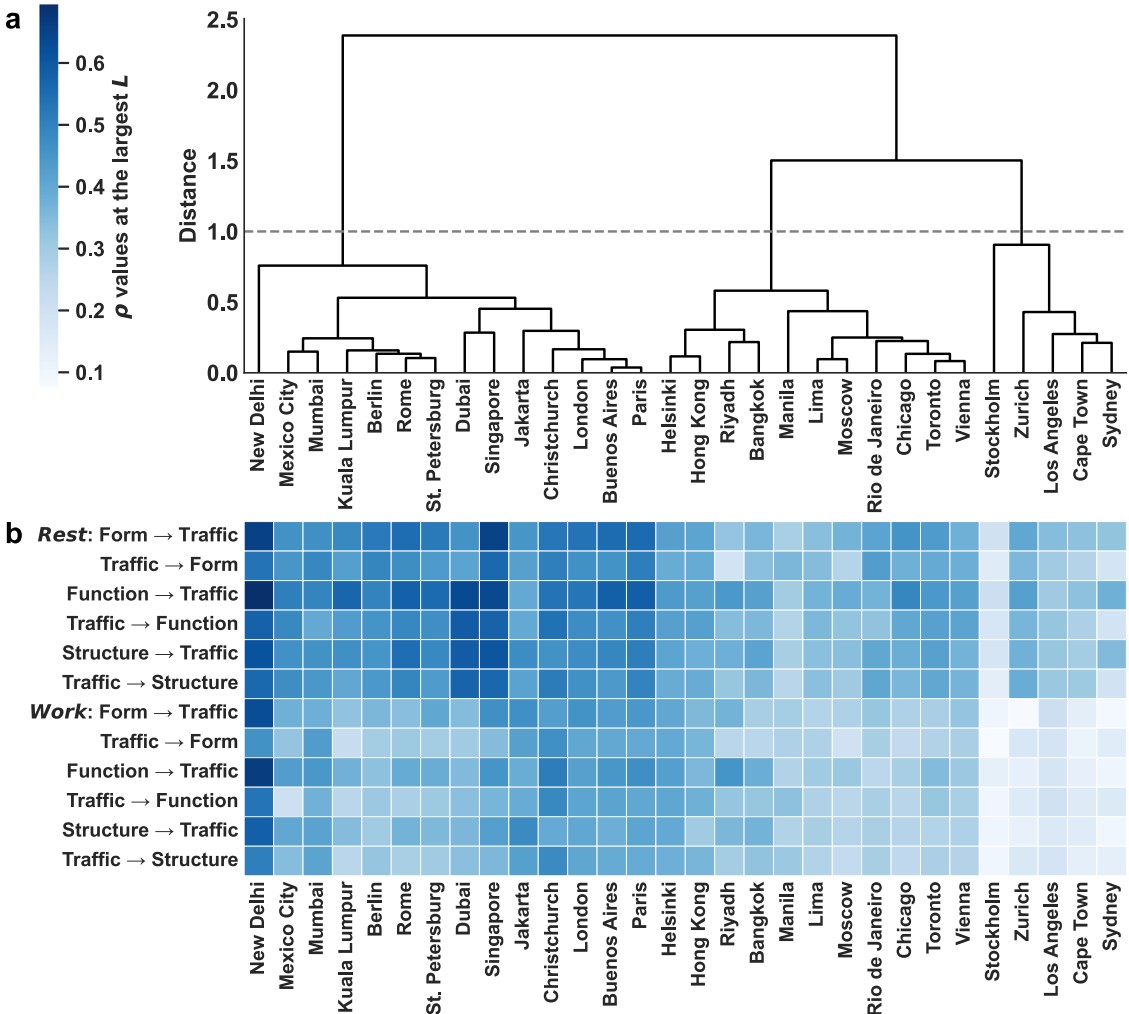

**Fig. 3 | Categorizing global cities through bidirectional causality between urban systems and traffic dynamics during rest and work days. a** Hierarchical clustering dendrogram of 30 cities based on the shape and magnitude of their bidirectional causality curves. The *x*-axis lists cities, and the *y*-axis shows the dissimilarity between their causal profiles. **b** Heatmap of average $\rho$ values at the largest library size $L$, estimated for each city on rest and work days. Columns follow the city order in the dendrogram. Rows indicate causal directions between urban systems and traffic dynamics, with the first six for rest days and the next six for work days.

## Discussion

While correlation and causation are intrinsically related, they remain fundamentally distinct in both implications and interpretability. Observed correlations do not necessarily imply direct influence, as they may arise from omitted variables or confounding factors, such as socioeconomic conditions and spatio-temporal dependencies in the urban environment[33,47]. By contrast, causality captures not only the strength but also the (often asymmetric) direction of influence, a distinction that is crucial for real-world applications. This distinction is important in sustainable urban development and intelligent transportation systems, where sound decision-making depends on identifying causal relationships rather than merely detecting statistical associations[39,48,49]. Accordingly, correlations alone are insufficient to justify interventions and must be complemented by causal inference strategies. For example, Stockholm shows consistently high $R^2$ values (>0.90) across all three urban system components, yet its $\rho$ values are among the lowest across the 30 cities. In contrast, New Delhi exhibits relatively lower $R^2$ values but among the highest $\rho$ values at the largest library size, indicating a stronger causal influence. Uncovering these directional causal effects offers deeper insights into the complex interplay between transportation infrastructure, mobility patterns,

and urban planning, ultimately enabling more effective data-driven policy interventions.

In this study, we find that urban structure often exhibits the strongest correlation with traffic dynamics, as observed in cities such as Singapore, New Delhi, London, Chicago, Moscow, and many other cities, largely because vehicular mobility tends to follow the layout and characteristics of the road network. The spatio-temporal causality framework, however, reveals that urban form and function more frequently drive the directional influence on congestion across these cities. In Singapore, for instance, the STWR model yields an $R^2$ of 0.95 for urban structure on rest days, substantially higher than the values for urban form (0.75) and function (0.89). Yet, the corresponding $\rho$ values at the largest library size $L$, reflecting the causal direction from urban systems to traffic dynamics, are 0.60 for structure, 0.65 for form, and 0.63 for function. This phenomenon aligns with a demand-side explanation: the spatial distribution of residences, workplaces, and activities, which are closely tied to urban form and function, generates and schedules trips that subsequently place pressure on the road network. Consequently, policy-makers should avoid relying solely on supply-side solutions confined to the road network; sustainable congestion mitigation requires coupling network operations with

demand-management strategies and land-use interventions that account for traveller behavior and spatial context[44,46].

On a global scale, our clustering does more than classify cities; it reveals three distinct causal archetypes that point to different pathways from diagnosis to intervention, laying the groundwork for city-to-city learning and knowledge exchange. In cities where urban systems and traffic dynamics are tightly coupled, integrated land-use and transport strategies appear particularly effective in channeling urban system signals toward congestion mitigation[37,43]. Where bidirectional influence persists but underlying patterns are heterogeneous, effective implementation requires a diagnostic-to-portfolio approach: use causal signatures to target corridors with network operations and districts with land-use and demand measures, instead of relying on uniform policy templates. By contrast, cities with attenuated causal patterns on workdays reveal the limitations of static levers and highlight the need for workday-focused operations combined with demand-side governance. Importantly, similar cluster membership does not imply identical outcomes, as socioeconomic conditions, spatial scales, and temporal rhythms can moderate causal pathways and confound transfer[3,31,33,44,49]. Thus, the value of our clustering lies less in the typologies themselves than in the actionable pathways they enable, which support context-sensitive interventions across Global North and South settings while remaining responsive to equity and place-specific dynamics[36,38,50].

In this context, it is important to note that our STCCM causal fingerprints aggregate local causal diagnostics, computed over within-city spatial units, into city-level $L$-$\rho$ patterns. For cities with strongly heterogeneous sub-regions, such as pronounced core-periphery contrasts or polycentric structures, the city-level curves may reflect a mixture of distinct local regimes rather than a single dominant mechanism. In these instances, weaker separations between bidirectional cross-mapping skills or less pronounced convergence should be interpreted cautiously, as they may arise from offsetting local causal pathways. This motivates future stratified extensions in which STCCM is applied separately across spatial-unit groups defined by congestion intensity, centrality, or urban form archetypes. Such an approach would better reveal sub-regional causal heterogeneity while maintaining cross-city comparability through a consistent spatial support and kernel design.

Our empirical findings are broadly consistent with, yet also extend, previous evidence on the links between urban systems and traffic dynamics[18–20,24,25]. By moving from correlations to explicit causal inference, our spatio-temporal causality framework complements this literature in two ways. Conceptually, a large share of existing work relies on global or locally varying associations[15,22], which are valuable for explanation and prediction but do not directly resolve directionality or feedback. In contrast, the STCCM model provides bidirectional and potentially asymmetric causal diagnostics, enabling a comparable assessment of how urban structure, form, and function exert directional influence on traffic dynamics across cities and between rest and work days. Empirically, the resulting causal fingerprints reveal recurring causal patterns and cross-city regularities, while also highlighting substantial context dependence. These findings suggest that interventions focusing solely on network supply may overlook critical demand-side mechanisms embedded in urban form and function. Consequently, effective mitigation requires a synergistic approach that coordinates land-use configuration and activity distribution with network operations, tailored to local causal pathways.

While this study provides a cross-city diagnosis of bidirectional and asymmetric causality between urban systems and traffic dynamics, several limitations warrant caution and point to directions for future research. First, the analysis relies on a road-based congestion metric and thus captures vehicular mobility rather than the full spectrum of human movement. Future research should incorporate multimodal data sources, such as pedestrian counts, public transportation smart card records, cycling sensors, and GNSS traces, to better represent multimodal trip chains and diverse activity patterns. Extending the temporal scope to cover seasonal variation, holidays, and policy shocks, and expanding the city sample beyond 30, will mitigate seasonal bias, improve representation across Global North and South contexts, and enhance external validity. Second, the composite indicators for each urban system component are constructed from a fixed set of globally available features to ensure cross-city comparability. While this design facilitates consistent benchmarking, it may underrepresent context-specific determinants of congestion in some cities (e.g., multimodal supply, demand-management policies, or informal transport), which can lower the explanatory power of the association models even when strong urban-traffic coupling exists. Future work could expand the feature set and explore hierarchical specifications or models tailored to specific city clusters to obtain more representative composite indicators across diverse urban contexts. Third, the practical value of causal signatures depends on their translation into actionable strategies. While our spatio-temporal causality framework reveals bidirectional interactions between urban systems and traffic dynamics, converting these diagnostics to practice requires feature-level targeting and integrated intervention portfolios. A factor-specific implementation of STCCM (applied to individual road networks, morphological properties, and land-use attributes rather than aggregate indices) can help identify component-dominant corridors and districts. In addition, incorporating external knowledge, such as common-sense reasoning and causal relationship extraction from natural language processing, may complement data-driven inference and strengthen the robustness of causal determination. These causal fingerprints can then be evaluated in digital-twin environments to simulate and optimize context-specific combinations of network operations, demand management, and place-based land-use interventions prior to real-world deployment. Overall, our results provide a scalable blueprint for translating causal diagnostics into targeted and durable congestion mitigation strategies aligned with broader sustainability objectives.

## Methods

### Screening cities and datasets for global reach

To thoroughly investigate how urban systems and traffic dynamics interact, it is crucial to select global cities representing diverse economic, cultural, and administrative contexts. Alongside the rapid urbanization observed worldwide, numerous cities have transitioned beyond their original administrative or industrial functions[4,51]. Existing studies and international organizations have attempted to classify global cities based on their developmental trajectories, functional characteristics, or international connectivity. Examples include distinguishing between the Global South and North and delineating multi-tier city-regions worldwide[35]. Based on this, we use three specific criteria to select the cities for exploring their bidirectional causality between urban systems and traffic dynamics: (1) data availability, (2) the extent of traffic congestion, and (3) geographical distribution to ensure global reach.

The first criterion is whether the necessary data sources are accessible. This study mainly relies on three datasets, i.e., traffic datasets for computing traffic dynamics indicators, OpenStreetMap (OSM) datasets for extracting urban system features, and global administrative boundaries (GADM) for defining city limits. In detail, traffic datasets from HERE Technologies provide traffic information in the form of jam factors for road segments, enabling the computation of mobility indicators across global cities[52,53]. OSM offers detailed datasets on transportation networks, building coverage, points of interest, and land use to derive urban system features[54]. GADM supplies boundary information to delineate the geographical extents of cities. Although OSM and GADM offer comprehensive global coverage, HERE Technologies imposes certain limitations on cities with available

traffic data. Consequently, the initial selection process prioritizes cities where coverage from HERE Technologies is confirmed.

After data availability is confirmed, the second criterion evaluates the congestion levels of traffic dynamics to ensure the representation of diverse human mobility patterns. Drawing on publicly available congestion reports from TomTom's 2023 traffic index[55], we incorporate both highly congested cities, such as Manila and Lima, in their broader metropolitan areas, as well as London and Toronto in more compact city centers, along with relatively uncongested cities, including Singapore, Zurich, and Stockholm. This selection guarantees a wide spectrum of traffic conditions across diverse transportation and infrastructural settings.

The third criterion emphasizes worldwide representation. To ensure that the findings reflect diverse political and economic contexts, the selected cities should span multiple continents and include major economic centers or national capitals, such as Los Angeles, Moscow, New Delhi, and Cape Town. This global perspective allows for examining spatial causality under varying governance structures, cultural landscapes, and developmental stages.

These three considerations lead to a final selection of 30 cities, ranging from prominent metropolitan capitals in developed nations to mid-sized economic centers in emerging economies. In this study, traffic datasets from HERE Technologies were obtained at 5-min intervals in two 1-month phases: June 1–30, 2024, and September 21–October 20, 2024, owing to data availability. A detailed visualization of their road networks extracted from HERE Technologies and administrative boundaries is presented in Fig. 4.

## Delineating traffic dynamics

Identifying appropriate spatial units is critical in delineating residents' mobility in the urban environment. The commonly used methods include grid-based divisions, traffic analysis zones (TAZs), Voronoi polygons, and buffer areas surrounding road segments[5,56]. Each method offers distinct advantages while also facing specific limitations. For example, grid-based divisions are straightforward to implement, but they often fail to align with functional boundaries. Voronoi polygons perform well in dense networks by capturing localized traffic dynamics, yet in sparsely populated areas, they tend to generate overly large and less informative units. TAZs capture broader environmental contexts due to their design around traffic flow, but lack fine resolution and strong connections to specific road segments. Buffer areas face challenges with multi-level disparities in buffer sizes across different road segments.

To address these limitations, we employ a hybrid approach that combines Voronoi-based divisions with buffer distances, resulting in Buffered Voronoi Cells (BVCells). The construction of BVCells begins with selecting intersection nodes within the road network. Intersection nodes are chosen as anchor points as they represent critical locations where traffic jams originate, such as those influenced by traffic light controls or congestion spillover effects[57]. However, many road intersections feature multiple nodes in close proximity, which can lead to an excessively dense computational unit when generating Voronoi polygons. To mitigate this issue, we apply a density-based clustering algorithm to merge nodes within a 20-m radius. Once the intersection nodes are merged, Voronoi polygons can be generated around these points. A buffer boundary is then applied to limit each polygon with a buffer size of 500 m[5], capturing the surrounding environment relevant to local traffic systems. This process also addresses the issue of sparsely located nodes generating disproportionately large units. A visualization of the BVCell division across the 30 selected cities is presented in Fig. 4.

For each BVCell, we compute a traffic dynamics indicator (TDI) to capture traffic dynamics over time. This indicator is derived from the real-time traffic dataset provided by HERE Technologies, a data source

shown to be reliable for measuring and forecasting traffic dynamics across multiple cities[52,53,58]. Supplementary Table 8 summarizes the dataset for all 30 cities and reports its spatial and temporal representativeness. The traffic dataset was collected every 5 min during two 1-month windows, either 1 Jun–30 Jun 2024 or 21 Sept–20 Oct 2024. For each city, the time-coverage ratio refers to the proportion of 5-min slots that contain valid traffic records in the 1-month window. We present a minimum of 98.40% and a median of 98.60% for all time-coverage ratios, showing high temporal representativeness with near-continuous coverage of both work and rest days. From the spatial perspective, the number of HERE road segments within each city's boundary ranges from 944 in Manila to 18,788 in Los Angeles. To evaluate spatial representativeness, we overlay a 1 km × 1 km grid on each city and compute the spatial-coverage ratio: for each grid cell that contains OSM roads for motor-vehicle traffic, i.e., classified as motorway, trunk, primary, secondary, tertiary, or their links, we check whether the grid cell also contains HERE road segments. The resulting spatial-coverage ratios exceed 70% in 28 out of 30 cities and reach about 95% in Buenos Aires, Sydney, and Paris. Although Mexico City and Riyadh achieve lower city-wide ratios, their central districts are still well covered by HERE road segments, as shown by the blue grids in the accompanying heatmaps. Overall, the HERE traffic dataset exhibits high spatial and temporal representativeness, supporting its use for TDI computation.

Given the traffic dataset, we use the jam factor, representing real-time traffic dynamics for each road segment, to compute the TDI. The jam factor is a dimensionless congestion index, ranging from 0.0 as free flow to 10.0 as blocked, which is obtained by applying a piecewise-linear mapping to the ratio between the current traffic speed and the expected default traffic speed[16]. Since each BVCell encompasses multiple road segments and time slots, we account for both spatial and temporal variability by applying separate weighting strategies. First, temporal weighting smooths the short-term fluctuations in traffic conditions over a month to generate the hourly traffic dynamics series. Each road segment $r$ is associated with a jam factor $\text{JF}_{(r, d, h, i)}$, where $d \in \{1, ..., 30\}$ is the day in the 1-month window, $h \in \{0, ..., 23\}$ is the hour of the day, and $i$ indexes the 5-min snapshots within each hour. We apply temporal weighting to aggregate jam factors in two steps, i.e., hourly aggregation and daily aggregation. We start with obtaining the hourly average $\text{JF}_{(r, d, h)}$:

$$\text{JF}_{(r, d, h)} = \frac{1}{m_{(d, h)}} \sum_{i=1}^{m_{(d, h)}} \text{JF}_{(r, d, h, i)}, \ 0 < m_{(d, h)} \leq 12, \tag{1}$$

where $m_{(d, h)}$ is the number of valid 5-min observations in hour $(d, h)$. To obtain a typical diurnal profile, we then average the hourly values across work and rest days separately:

$$\text{JF}_{(r, h)}^{\text{work}} = \frac{1}{D_{\text{work}}} \sum_{d \in \text{workdays}} \text{JF}_{(r, d, h)}, \ \text{JF}_{(r, h)}^{\text{rest}} = \frac{1}{D_{\text{rest}}} \sum_{d \in \text{restdays}} \text{JF}_{(r, d, h)}, \tag{2}$$

with $D_{\text{work}}$ and $D_{\text{rest}}$ denoting the numbers of work and rest days in the month, respectively. For simplicity, we denote either of them as $\text{JF}_r^t$. Second, spatial weighting recognizes the relative importance of each road segment within a BVCell by assigning weights proportional to its length. Given hour $t$, the TDI for BVCell $s$ is the length-weighted average of the hourly jam factor values of all road segments within $s$. Any segment that spans multiple BVCells is clipped at the cell boundaries. Let $R_s$ represent the set of road segments in BVCell $s$. Then, the spatially weighted TDI is defined as:

$$\text{TDI}_s^t = \frac{\sum_{r \in R_s} \left( \text{JF}_r^t \times L_r \right)}{\sum_{r \in R_s} L_r}, \tag{3}$$

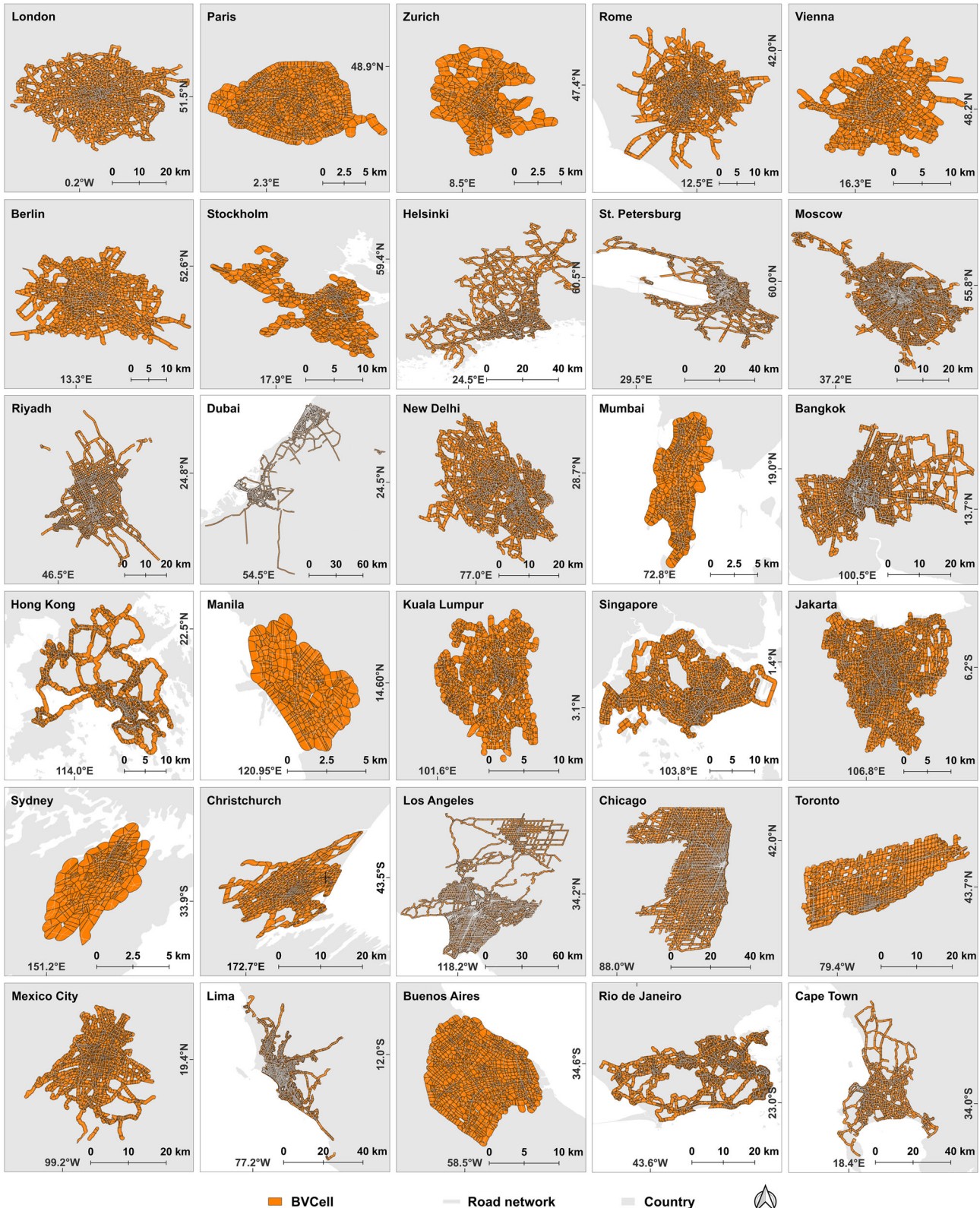

**Fig. 4 | Road networks extracted from HERE Technologies and their Buffered Voronoi cells (BVCell) across 30 global cities, including London, Paris, Zurich, Rome, Vienna, Berlin, Stockholm, Helsinki, St. Petersburg, and Moscow in Europe; Riyadh, Dubai, New Delhi, Mumbai, Bangkok, Hong Kong, Manila, Kuala Lumpur, Singapore, and Jakarta in Asia; Sydney and Christchurch in** Oceania; Los Angeles, Chicago, Toronto, and Mexico City in North America; **Lima, Buenos Aires, and Rio de Janeiro in South America; and Cape Town in Africa.** Administrative boundaries and road networks are derived from the Global Administrative Boundaries (GADM) and HERE Technologies, respectively.

where $\text{TDI}_s^t$ denotes the TDI value for BVCell $s$ at hour $t$ and $L_r$ is its length. The same calculation is performed separately for work and rest days, yielding two hourly TDI series for each BVCell.

## Delineating urban systems

Urban systems are complex and can be characterized by three interconnected components: structure, form, and function[8,13,59]. These components constitute the "triple helix" of urban systems, involving their physical, morphological, and functional dimensions. Note that this differs from the "triple helix" model, often referred to as describing university-industry-government relations in innovation systems[60], as our framework focuses specifically on the interplay among structure, form, and function of the urban environment. Here, urban structure constitutes the physical framework and network backbone of a city, primarily shaped by its transportation networks[9]. Then, urban form captures its morphological characteristics, such as spatial layouts, while urban function describes the roles and utilization of these spatial configurations[11]. To ensure cross-city comparability, we compute these urban system features from globally available OSM data and select variables based on three criteria: (i) representing complementary aspects of the three components; (ii) established interpretability in existing urban and transport studies; and (iii) consistent computability at the BVCell scale across cities.

Urban structure is a multifaceted concept, encompassing polycentricity, topological organization of road networks, and other related aspects[61,62]. This study focuses on the spatial structure of road networks derived from OSM data as the foundation for calculating topological features. To comprehensively characterize the road networks within each BVCell, we define urban structure metrics from three perspectives: (1) node-related metrics quantify the properties of intersections and junctions in the network, including node count (NODE_CT), node density (NODE_DENS), crossing count (CRS_CT), and average node degree (K_AVG); (2) edge-related metrics depict the properties of road segments in the network, including total length (EDGE_LEN), road density (RD_DENS), and network intensity (NET_INT); and (3) centrality-related metrics assess the importance and connectivity of road segments in the network, including average betweenness centrality (BTW_CEN), average closeness centrality (CLS_CEN), and average degree centrality (DEG_CEN).

Urban form defines the spatial layout and arrangements of the built environment, commonly measured through morphological metrics[11,63]. This study employs a comprehensive set of metrics at the landscape level to delineate urban form for each BVCell, categorized into four dimensions: (1) area-edge metrics quantify the size and edge characteristics of spatial units, including total area (TA), number of patches (NP), patch density (PD), total edge (TE), and edge density (ED); (2) shape metrics capture shape properties from the geometric perspective, such as landscape shape index (LSI), perimeter-area ratio (PARA), shape index (SHAPE), and fractal dimension (FRAC); (3) aggregation metrics describe the spatial cohesion or dispersion of patches, including contagion (CONTAG), aggregation index (AI), cohesion (COHES), and splitting index (SPLIT); and (4) diversity metrics reflect the variety and evenness of spatial patterns, using measures such as patch richness (PR), Shannon's diversity index (SHDI), and Shannon's evenness index (SHEI).

Urban function represents the socioeconomic and land-use attributes of urban space[64]. In this study, the functional features of each BVCell are quantified using the area percentage of various land-use types. The land-use classification system is established based on OSM datasets through a hierarchical computational framework[65], with modifications to better align with human mobility patterns. The following land-use categories are used to depict urban functions: transportation area (TRNS), commercial and business facilities area (COMM), industrial area (IND), residential area (RES), farmland and forest area (FRST), education and science area (EDU), sports area

(SPRT), park area (PARK), recreation area (RECR), water area (WATR), health care area (HLTH), cultural facilities area (CULT), and administration area (ADMIN).

Descriptions of the features characterizing urban structure, form, and function are provided in Supplementary Table 9. Interdependent by nature, these three dimensions represent distinct facets of urban systems, each influencing human mobility in different yet complementary ways. Essentially, urban structure provides the spatial framework that enables connectivity and accessibility, while urban form and function co-evolve to shape activity distribution and influence travel behavior. By differentiating these components, we can more comprehensively disentangle their unique contributions to traffic dynamics.

## Spatio-temporal causality framework

We propose a spatio-temporal causality framework to investigate the bidirectional causal patterns between urban systems and traffic dynamics, comprising two complementary components: spatio-temporal weighted regression (STWR) and spatio-temporal convergent cross mapping (STCCM). The first step involves applying the STWR model to quantify the relationship between traffic dynamics and urban structure, form, and function. Throughout this process, the STWR model accounts for temporal variations while generating geographically weighted coefficients for urban system features at each location. To integrate insights from multiple metrics within each component of urban systems, such as the 16 metrics representing urban form, their geographically weighted coefficients are utilized to construct composite indicators for urban structure, form, and function, respectively. Then, the composite indicators serve as input to the STCCM model, which explores the bidirectional causal patterns between urban systems and traffic dynamics during congestion periods. Following this, a clustering approach grounded in bidirectional causality synthesizes the three urban system components to uncover patterns of similarity and differences across global cities.

**Spatio-temporal weighted regression.** To address spatial non-stationarity in geographical processes, the geographically weighted regression (GWR) model was introduced to account for spatial variability by estimating local parameters[66]. However, it struggles with temporal variations in dynamic systems, which prompted the development of spatio-temporal extensions that capture evolving patterns over space and time, such as spatio-temporal weighted regression (STWR) and geographically and temporally weighted regression (GTWR)[21,67]. Here, we adopt STWR for its ability to update spatio-temporal kernel functions and adjust temporal bandwidths dynamically, thereby better capturing complex spatio-temporal interactions and aligning with the rapidly changing nature of traffic dynamics.

When examining the relationship between urban systems and traffic dynamics, urban structure, form, and function features are assumed to remain constant over the 1-month study period, as their temporal variations are expected to be negligible. However, the relationship between these static features and traffic dynamics can vary over time, capturing dynamic patterns inherent to traffic systems. To address multicollinearity, we employ the variance inflation factor (VIF) to iteratively identify and remove urban system features with VIF values above 10 before applying the STWR model (see Supplementary Fig. 3)[68]. Given $k$ dependent variables from each component of urban systems, the formulation of STWR can be expressed as:

$$y(s,t) = \beta_0(s,t) + \sum_{k=1}^{p} \beta_k(s,t)x_k(s) + \varepsilon(s,t), \tag{4}$$

where $y(s,t)$ is the dependent variable TDI for BVCell $s$ at time $t$, $x_k(s)$ are the $k$ independent variables for BVCell $s$, $\beta_k(s,t)$ are the coefficients that vary across both space and time, and $\varepsilon(s,t)$ is the error term.

During the estimation process, we employ a spatio-temporal bi-square kernel function to adaptively assign weights based on spatial and temporal proximity[67]. In addition, when implementing the STWR model, we include TDI from 06:00 to 23:00 to account for active human behavior while excluding non-service hours of urban infrastructure[21]. The performance comparisons of GWR and STWR are reported in Supplementary Tables 3 and 4 for rest and work days, respectively.

After deriving the spatially and temporally weighted coefficients from the STWR model, a bandwidth-based smoothing approach was employed to construct composite indicators for each component of urban systems. Given that $\beta_k(s, t)$ varies across spatial and temporal dimensions, a bi-square kernel function was applied to derive the smoothed coefficient $\widetilde{\beta}_k(s, t)$ for feature $k$ at location $s$ and time $t$. Using the smoothed coefficients, the composite indicator for each BVCell was computed as a weighted sum of the input variables, incorporating both the intercept term and the contributions of each feature. The composite indicator CI($s, t$) for BVCell $s$ at time $t$ was computed as:

$$CI(s, t) = \beta_0(s, t) + \sum_{k=1}^{p} \widetilde{\beta}_k(s, t) x_k(s), \tag{5}$$

where $x_k(s)$ represents the standardized values of the independent variables. This composite indicator captures the aggregated impact of each urban system component on traffic dynamics, which retains critical spatial heterogeneity while reflecting the localized interactions between urban systems and mobility patterns. However, since these composite indicators are derived from a globally consistent feature set, cross-city differences in STWR model fit may reflect not only heterogeneous urban-traffic interactions but also the degree to which the selected variables capture locally salient determinants of traffic patterns.

In urban systems, strict stationarity is rarely plausible due to the inherent spatial heterogeneity and temporal non-stationarity of interactions between the built environment and traffic dynamics[21,22,66]. Consequently, our framework relaxes the stationarity requirement by first estimating spatially and temporally varying coefficients with STWR and then smoothing these coefficients with a bi-square kernel to construct composite indicators $CI(s, t)$[67,69]. This procedure yields quasi-stationary neighborhoods in which the association structure is approximately stable within the local support defined by the STWR kernel and the BVCell aggregation. Practically, the STWR bandwidth controls the bias-variance trade-off: narrower bandwidths preserve fine-grained heterogeneity but may retain more local noise, while wider bandwidths improve stability but may blend distinct local regimes. In addition, the BVCell buffer size determines the spatial support over which traffic and surrounding urban contexts are aggregated, thereby affecting the homogeneity of each cell. The choice of a 500-m buffer is consistent with prior evidence on the scale of local traffic impacts[5,16]. Increasing this size would likely increase signal stability through aggregation but decrease the sensitivity to micro-scale urban system variations.

**Spatio-temporal convergent cross mapping.** Establishing causation, rather than merely identifying correlation, determines whether and how changes in one variable drive changes in another. So far, many methods have been developed to address this issue, such as structural causal models and causal graphical models[70]. Among them, convergent cross mapping (CCM) stands out for its capacity to infer both the presence and direction of causality in complex dynamical systems[71]. Building on CCM, the geographical convergent cross mapping (GCCM) model was introduced for spatial causal inference by utilizing spatial cross-sectional data and reconstructing state spaces[34]. Although GCCM effectively integrates geographical information, the

lack of temporal dynamics makes it less suited for rapidly changing processes, such as traffic dynamics. To address this limitation and align with the STWR model, we propose a customized temporal period selection model, i.e., spatio-temporal convergent cross mapping (STCCM), to capture the bidirectional causal pattern between urban systems and traffic dynamics under time-varying conditions.

STCCM builds upon the STWR-based composite indicator CI($s, t$) to represent each component of urban systems, denoted as $X(s, t)$ for BVCell $s$ at time $t$, while the TDI is treated as $Y(s, t)$. The objective of STCCM is to determine whether $X$ can reconstruct $Y$ and vice versa, thus elucidating the direction and strength of causal influences. Following Takens' embedding theorem, the shadow manifolds $M_X$ and $M_Y$ can be reconstructed from $X$ and $Y$ to preserve the topological equivalence of the original dynamical system[30]. Cross-mapping then leverages the correspondence between these shadow manifolds to infer causality. If $X$ causes $Y$, information about $X$ is encoded in $M_Y$, enabling predictions of $X$ using $M_Y$; conversely, $Y$ causing $X$ allows predictions of $Y$ using $M_X$. In detail, the STCCM model is implemented in three steps: state space reconstruction, simplex projection, and bidirectional cross mapping.

State space reconstruction. We first reconstruct the state spaces $M_X$ and $M_Y$ based on $X$ and $Y$, respectively. Recognizing that congestion periods vary across locations and are spatially propagated[57], we focus on the most congested period for each BVCell. This ensures that the reconstructed state space captures local spatial interactions under specific temporal conditions. In addition, as traffic patterns differ significantly between workdays and rest days, we separately consider these two scenarios. For BVCell $s$, we identify its most congested time $t_s$ and reconstruct its state space under this period. Given an embedding dimension $E$, the embedding vector for $\mathbf{X}(s, t_s)$ is defined as:

$$\mathbf{X}(s, t_s) = [x(s, t_s), x(s_1, t_s), \ldots, x(s_{E-1}, t_s)], \tag{6}$$

where $x(s_i, t_s)$ represents the value of a lagged spatial unit $s_i$ with order $i$ at time $t_s$. Here, the first-order spatial lag includes all adjacent BVCells to $s$, while second-order lags are derived from the first-order neighbors, and so forth[34]. Thus, the embedding dimension $E$ determines how many spatial-lagged neighbors are included in the reconstructed state space, where a larger $E$ corresponds to higher-order spatial neighborhoods. The sensitivity analysis of STCCM with varying $E$ is reported in Supplementary Figs. 4 and 5. Figure 5 provides a schematic illustration of spatial lags across different orders and time periods. Similarly, the embedding vector for $\mathbf{Y}(s, t_s)$ that encapsulates local traffic patterns at time $t_s$ is constructed as:

$$\mathbf{Y}(s, t_s) = [y(s, t_s), y(s_1, t_s), \ldots, y(s_{E-1}, t_s)]. \tag{7}$$

Since each order of the spatial lag encompasses multiple BVCells, we apply a weighted averaging method to aggregate CI and TDI values across all BVCells within the order. The construction of $x(s_i, t_s)$ employs the BVCell area as the weighting coefficient for CI, while the road length within each BVCell serves as the weighting coefficient for TDI when constructing $y(s_i, t_s)$.

Simplex projection. Once the embeddings in the reconstructed state space are generated, STCCM employs a simplex projection scheme to quantify the causal influence between $\mathbf{X}(s, t_s)$ and $\mathbf{Y}(s, t_s)$. Simplex projection estimates the target variable by locating its nearest reconstructed states and forming a distance-weighted average of their observed values. For BVCell $s$, we use a specific library size $L$ to define a library set of BVCells, which are determined based on spatial proximity to $s$. Then, given $x(s, t_s)$, the value of $y(s, t_s)$ can be predicted according to its nearest neighbors identified from $M_X$ in this library set. Here, we set the number of nearest neighbors as $E + 1$[71]. Each neighbor contributes to the prediction through a weighted average of its corresponding target values, where the weights decrease with distance to

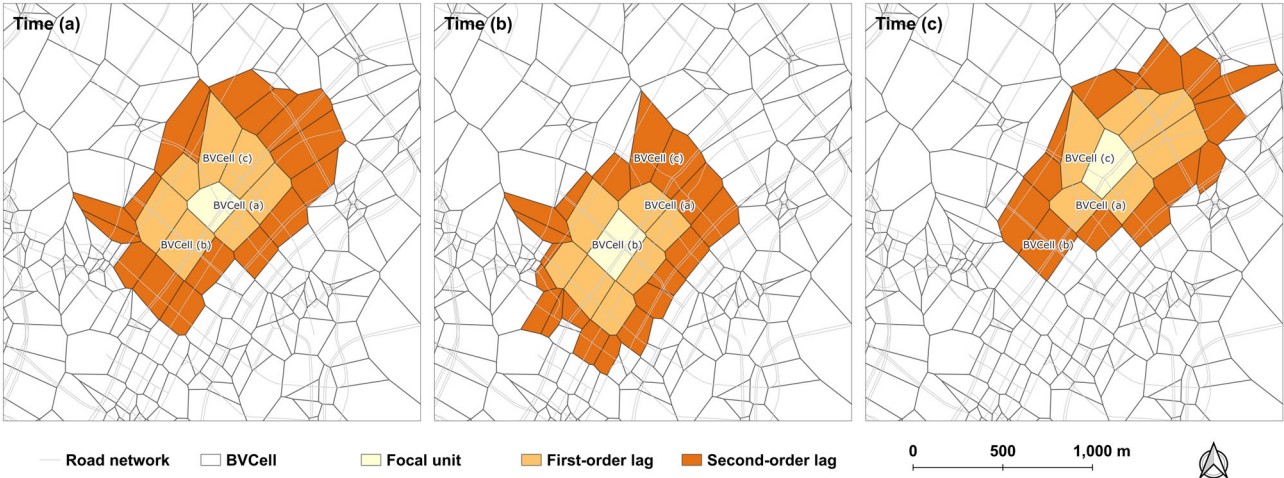

**Fig. 5 | Schematic illustration of spatial lags across different orders and time periods.** The focal unit corresponds to the Buffered Voronoi cell (BVCell) used for state-space reconstruction, with first-order lags defined by adjacent BVCells, second-order by the neighbors of the first-order lags, and higher orders iteratively derived. For each BVCell, the state space is reconstructed based on its most congested time, denoted as $t_s$. For example, Time (**a**) corresponds to the most congested period for BVCell (**a**), Time (**b**) for BVCell (**b**), and Time (**c**) for BVCell (**c**). Administrative boundaries and road networks are derived from the Global Administrative Boundaries (GADM) and HERE Technologies, respectively.

prioritize closer neighbors. The predicted value for a target BVCell $s$ at time $t_s$ is computed as:

$$\hat{y}(s, t_s)|M_X = \frac{\sum_{i=1}^{E+1} w_{si} y(s_i, t_s)}{\sum_{i=1}^{E+1} w_{si}}, \tag{8}$$

where $w_{si}$ is the weight for the $i$-th nearest neighbor of $s$[34].

Bidirectional cross-mapping. To assess bidirectional causality, we can reverse the roles of $X$ and $Y$ in cross-mapping. The cross-mapping skill is quantified by the Pearson correlation coefficient $\rho$ between the observed and the predicted values. A higher $\rho$ indicates greater predictive skill and serves as a measure of causal strength. Specifically, $X \to Y$ uses $X$ as the predictor variable to reconstruct the state space and predict $Y$, whereas $Y \to X$ uses $Y$ as the predictor variable to reconstruct the state space and predict $X$. We vary the library size $L$ in increments of 5, ranging from 5 to 500, to observe how $\rho$ evolves as more BVCells are incorporated. For cities with a maximum library size smaller than 500, we use the largest library size permissible by the data, such as Manila. The causal influence is determined by comparing the cross-mapping skills $\rho_{X \to Y}$ and $\rho_{Y \to X}$. If $\rho_{X \to Y}$ converges with increasing library size and surpasses $\rho_{Y \to X}$, $X$ is inferred to have a dominant causal influence on $Y$. Conversely, if $\rho_{Y \to X}$ is higher, $Y$ is inferred to dominate the causal relationship.

Since STCCM inputs are derived from locally fitted and smoothed STWR models, key scale parameters can influence the convergence behavior of the cross-mapping skill $\rho$ with increasing library size $L$. Intuitively, a narrower STWR bandwidth (or finer BVCell partition) yields more localized and potentially more heterogeneous composite-indicator fields[67,69], which can reduce the effective signal-to-noise ratio and lead to slower or less stable convergence in $L$-$\rho$ curves. Conversely, overly large bandwidths (or coarse spatial aggregation) may blur distinct sub-regimes and attenuate causality signals by mixing mechanisms, producing flatter $L$-$\rho$ curves and weaker separations between $\rho_{X \to Y}$ and $\rho_{Y \to X}$. Therefore, the selected bandwidth and BVCell buffer size should be interpreted as defining the operative scale of quasi-stationarity, within which the local causal diagnostics are valid.

To assess parameter robustness, we conducted a sensitivity analysis of the STCCM model by varying $E$ at the largest $L$ for both rest and work days across 30 cities, with results reported in Supplementary Figs. 4 and 5. In addition, we performed statistical significance tests on the cross-mapping skill derived from the STCCM model. As our analysis covers the full study area partitioned into BVCells, we utilize the distribution across these spatial units as a standardized basis for statistical testing. First, the statistical significance of $\rho$ at each library size was evaluated using a one-sample Student's $t$-test across all BVCells (see Supplementary Tables 5 and 6). Second, to verify the observed asymmetry, we assessed the significance of the difference between directional causal strengths ($\Delta\rho = \rho_{X \to Y} - \rho_{Y \to X}$) at the largest library size. This was conducted using a paired Student's $t$-test comparing the distributions of causal strengths in both directions across all BVCells for each city (see Supplementary Table 7). All tests reported are two-sided with a significance threshold of 0.05.

**Bidirectional causality clustering for global cities.** After deriving the bidirectional causal patterns between urban systems and traffic dynamics, we categorize global cities to explore their similarities and differences by analyzing the shape characteristics of the $\rho - L$ curves. We employ two distinct distance measures to characterize the $\rho - L$ curves for urban systems and their causal relationships with traffic dynamics across both work and rest days. The first measure, dynamic time warping (DTW), assesses the similarity of $\rho - L$ curves by nonlinearly aligning them to capture shape variations in bidirectional causal patterns[72]. The second measure, the Jaccard index, quantifies the overlap of areas under the curves to reflect the overall magnitude and distribution of causality influences across cities[73].

Given the bidirectional causality results from STCCM for all library sizes within a city $m$, the two $\rho - L$ curves are denoted as $P_{Y \to X}^{m}$ and $P_{X \to Y}^{m}$. The DTW distance $\mathrm{DTW}_{(m, n, r)}$ between cities $m$ and $n$ for urban systems $X \in \{X_{\mathrm{str}}, X_{\mathrm{frm}}, X_{\mathrm{fun}}\}$ and traffic dynamics $Y$ on rest days $r$ is computed as follows:

$$\mathrm{DTW}_{(m, n, r)} = \sum_{X \in \{X_{\mathrm{str}}, X_{\mathrm{frm}}, X_{\mathrm{fun}}\}} \left[ \mathrm{DTW}(P_{X \to Y}^{(m, r)}, P_{X \to Y}^{(n, r)}) + \mathrm{DTW}(P_{Y \to X}^{(m, r)}, P_{Y \to X}^{(n, r)}) \right], \tag{9}$$

where $X_{\mathrm{str}}$, $X_{\mathrm{frm}}$, and $X_{\mathrm{fun}}$ refer to structure, form, and function, respectively.

Similarly, we quantify magnitude differences using a Jaccard distance defined on regions under the bidirectional $L$-$\rho$ curves over $L \in [L_{\min}, L_{\max}]$, where $[L_{\min}, L_{\max}]$ denotes the library-size range considered in STCCM. For each city $m$ and dimension $X \in \{X_{\mathrm{str}}, X_{\mathrm{frm}}, X_{\mathrm{fun}}\}$

on rest days $r$, we define each directional region as the area between the curve and the $L$-axis (i.e., $u = 0$):

$$A_{X \to Y}^{(m,r)} = \{(L, u) : L \in [L_{\min}, L_{\max}], \; 0 \le u \le P_{X \to Y}^{(m,r)}(L)\}, \quad (10)$$

$$A_{Y \to X}^{(m,r)} = \{(L, u) : L \in [L_{\min}, L_{\max}], \; 0 \le u \le P_{Y \to X}^{(m,r)}(L)\}. \quad (11)$$

In practice, each directional region is represented as a polygon by connecting sampled points $(L_i, \rho(L_i))$ and closing the boundary with $u = 0$. The bidirectional region is then defined as

$$A_X^{(m,r)} = A_{X \to Y}^{(m,r)} \cup A_{Y \to X}^{(m,r)}. \quad (12)$$

Given two cities $m$ and $n$, the Jaccard index $J(\cdot)$ is computed as the intersection-over-union of these bidirectional regions:

$$J\left(A_X^{(m,r)}, A_X^{(n,r)}\right) = \frac{\left| A_X^{(m,r)} \cap A_X^{(n,r)} \right|}{\left| A_X^{(m,r)} \cup A_X^{(n,r)} \right|}, \quad (13)$$

and the Jaccard distance is $\text{JaccardDist} = 1 - J(\cdot)$. Accordingly, the overall Jaccard distance between cities $m$ and $n$ on rest days $r$ is computed as follows:

$$\text{Jaccard}_{(m,n,r)} = \sum_{X \in \{X_{\text{str}}, X_{\text{frm}}, X_{\text{fun}}\}} \text{JaccardDist}\left(A_X^{(m,r)}, A_X^{(n,r)}\right). \quad (14)$$

The overall distance $D^{(m,n)}$ between cities $m$ and $n$ is determined by a weighted combination of the normalized DTW and Jaccard distances, formulated as follows:

$$D_{(m,n)} = \lambda(\text{DTW}'_{(m,n,r)} + \text{DTW}'_{(m,n,w)}) + (1 - \lambda)(\text{Jaccard}'_{(m,n,r)} + \text{Jaccard}'_{(m,n,w)}), \quad (15)$$

where DTW$'$ and Jaccard$'$ represent the normalized DTW and Jaccard distances to ensure unit consistency. The parameter $\lambda$, set to 0.5, assigns equal weight to both measures. The subscripts $r$ and $w$ denote rest and work days, respectively. Then, hierarchical clustering is applied to the overall city-to-city distances $D_{(m,n)}$ to categorize global cities, enabling a multi-level exploration of urban similarities.

### Reporting summary

Further information on research design is available in the Nature Portfolio Reporting Summary linked to this article.

## Data availability

The traffic datasets used to compute traffic dynamics indicators can be downloaded through the HERE API (https://www.here.com/developer). OSM and GADM data, used for computing urban system metrics, can be publicly accessed from https://download.geofabrik.de/ and https://gadm.org/data.html, respectively. The urban and traffic feature data generated in this study have been deposited in Figshare: https://doi.org/10.6084/m9.figshare.28656800. Source data are provided with this paper.

## Code availability

All codes that support the findings of this study are available in Figshare via the following link: https://doi.org/10.6084/m9.figshare.28656800.

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

## Acknowledgements

The research was conducted at the Future Resilient Systems at the Singapore-ETH Centre, which was established collaboratively between ETH Zurich and the National Research Foundation Singapore. This research is supported by the National Research Foundation Singapore (NRF) under its Campus for Research Excellence and Technological Enterprise (CREATE) programme (Y.Z., M.R.).

## Author contributions

Y.Z. and M.R. conceived the study. Y.Z. developed the methodology, implemented the software, and wrote the original draft. Y.Z. and Y.H. conducted the analysis. All authors reviewed and edited the manuscript. M.R. and S.G. supervised the project.

## Funding

## Competing interests

The authors declare no competing interests.
