## [Transparent Peer Review file · Nature Communications]

Bidirectional yet asymmetric causality between urban systems and traffic dynamics in 30 cities worldwide

Corresponding Author: Dr Yatao Zhang

Version 0:

Reviewer comments:

Reviewer #1

(Remarks to the Author)

This paper introduces STCCM (spatio-temporal convergent cross mapping) to investigate bidirectional causal relationships between urban systems and traffic dynamics. The topic is timely and relevant: uncovering causal dependencies in urban processes remains a central yet underexplored challenge in urban science. The proposed approach has the potential to advance our understanding of how urban structure and traffic mutually influence each other. However, the presentation of the methodology and the results requires substantial improvement before the contribution can be properly assessed.

Detailed comments

1) Several key concepts are fully delegated to the Methods section. While technical details can indeed be confined there, the main text should still provide intuitive explanations of the central ideas. This is particularly true for STCCM, which represents the core methodological innovation of the paper. Without a conceptual introduction in the main text, readers cannot grasp the novelty or rationale of the method.

2) The discussion would benefit from stronger engagement with recent studies on the interplay between urban form and traffic dynamics (e.g., <https://arxiv.org/abs/2510.02582>, <https://onlinelibrary.wiley.com/doi/10.1155/2023/6144048>, <https://www.sciencedirect.com/science/article/abs/pii/S0966692321002532>

). These works provide useful benchmarks and could help position the proposed approach within the broader literature. The authors should systematically collect and discuss similar studies, emphasizing what STCCM adds conceptually or empirically.

3) Some figures are difficult to interpret (e.g., Fig. 1f and Fig. 2a). Displaying all cities on a world map adds little to the understanding of the results. It would be clearer to present the graphs or scatter plots as standalone figures, perhaps selecting a subset of representative cities. This would considerably improve readability and allow the reader to focus on the patterns conveyed by the plots.

4) The meaning of some features is not immediately intuitive. For instance, SPRT (sports area) is self-explanatory, but CONTAG (contagion) is not. The main text should briefly describe each feature the first time it is mentioned, so that the discussion can be followed without repeatedly consulting the Supplementary Information.

5) Significance of p values in L– p analysis (Fig. 2b–d). In Figs. 2b–d, most cities appear close to the diagonal, suggesting that the estimated p values may not be statistically significant. The authors should clarify whether a threshold for the significance of p exists, how it is computed, and whether the reported relationships pass such a test. This is an important aspect for assessing the robustness of the findings.

6) At line 202, the paper mentions the “Jaccard similarity” but does not specify what entities are being compared. This issue occurs in several instances: key technical aspects are mentioned without sufficient explanation, making the paper difficult to follow for readers unfamiliar with the specific computational pipeline.

In summary, the paper tackles an important and underexplored topic with a potentially valuable methodological contribution,

but the clarity, contextualization, and interpretability of both the methods and results must be significantly improved.

(Remarks on code availability)

The repository's README file describes the code structure, data organization, and system requirements necessary to replicate the study. Although I did not attempt a full replication, the repository appears to provide sufficient information and resources to enable reproducibility.

Reviewer #3

(Remarks to the Author)

The current landscape of urban simulations, predictive, and optimization models is largely fragmented, with a predominant presence of solutions tailored to a specific cities or regions. Significant differences in urban layouts and traffic dynamics hamper the easy transfer of models between the cities. In modern conditions, when the pace of change in city infrastructure and mobility patterns can be compared to the pace of model development itself, the ability to accelerate the process by exploiting and fine-tuning results from similar cities becomes crucial. How can such similarity be defined and empirically evaluated so that cities can be meaningfully partitioned into clusters with similar traffic dynamics, and, moreover, similar types of interventions to reduce congestion?

The work answers this question for the first time by considering not only a single aspect (e.g. topology of the road network) but the interplay between structure, form, function and dynamics. This makes it highly significant not only for urban studies and geospatial science, but also for Urban AI applications, suggesting a set of interpretable variables that can serve as a context which conditions decision-making policies and facilitates transfer learning between cities. The latter is also supported by the city causal archetypes revealed in the analysis, which enable the identification of cities with similar coupling between urban systems and traffic dynamics. A second aspect defining the significance of the study is its focus on causation rather than correlation. This provides a foundation for scientifically justified transportation policies and interventions, as well as for designing assumptions and induction biases for intelligent transportation models which both are highly relevant and challenging problems in the field of intelligent transportation systems. In summary, this work is highly significant for several research fields including urban studies, geospatial science, intelligent transportation systems and data-driven traffic simulation.

The work is grounded in well-established literature on causal inference, spatio-temporal prediction models, geospatial information science and urban mobility studies. The application of Convergent Cross Mapping (CCM) to detect (potentially bidirectional) causalities is methodologically sound. The original CCM method assumes that the underlying dynamical system is stationary, so that the functional relationships between observables remain constant over time. However, this assumption may not hold for the systems under study which are characterized by spatial heterogeneity and temporal variability. The authors relax the strict stationarity assumption (similarly to GCCM but extending to both spatial and temporal dimensions) and propose novel ST-CCM method for bidirectional causal estimation in spatio-temporal systems. ST-CCM combines spatio-temporal CCM with spatio-temporal weighted regression (STWR) that smooths the original data into quasi-stationary, piecewise regions. The approach appears both methodologically innovative and practically sound, as it denoises the data and transforms them into a set of interpretable composite indicators which tend to be more stationary across space-time cells than the original variables. Since the data points to calculate the estimates of ρ are derived from locally fitted STWR regression models, a brief discussion on the theoretical and practical implications of relaxing the stationarity assumption (e.g. how the bandwidth of the STWR kernel or buffer size of Voronoi cells influence the convergence of ST-CCM, or how to interpret the results for the cities comprising sub-regions with highly heterogeneous structure, form or function), would further strengthen the methodological contribution of the study.

In the study, STWR models are estimated for three groups of variables representing urban structure, form, and function, and a single response variable related to traffic congestion. This enables a comparative evaluation of how well congestion can be explained by the composite indicators across cities worldwide. The results demonstrate that the proposed indicators have strong and significant correlation with traffic dynamics, with urban structure contributing more than form or function. A short discussion of the variable selection for composite indicators would further strengthen this result. For example, if the chosen variables primarily reflect certain types of cities, the regression model may exhibit lower explanatory power not only because the underlying processes differ, but also because the feature set is less representative for other urban contexts.

The third important result of the study is the development of hierarchical clustering approach that groups the cities according to the direction and strength of causality between urban structure, form, and function, and traffic dynamics), using ST-CCM outputs (L- ρ curves) as input data. The clustering results reveal the existence of three distinct causal archetypes of cities. This finding is highly relevant for scientifically grounded design of transportation policies and interventions, as the identified direction and strength of causality in a city highlights specific groups of measures that could be the most effective in mitigating congestion.

The work supports all its conclusions and claims, and no additional evidence is needed. In particular, the goodness of fit of the STWR regression for modelling the traffic indicator is supported by consistently high R^2 and AICc values compared to the GWR model. The authors evaluate the significance of ρ values for two library sizes, which aligns with standard CCM validation practice (see, e.g., Gao, B., Yang, J., Chen, Z. et al. Causal inference from cross-sectional earth system data with geographical convergent cross mapping. *Nat Commun* 14, 5875 (2023)). The general conclusions regarding the causal interplay between urban systems and traffic dynamics, as well as the explanatory power of the urban structure, form, and function composite indicators, are well supported by the extensive dataset covering 30 cities on multiple continents. Moreover, the uniformity of the traffic congestion data which are sourced from the same provider and tested for spatial and

temporal representativeness, adds credibility to the reported findings.

The study provides enough details in the methods to ensure reproducibility. As a minor suggestion, in Figure 1c-e, the zero reference line in the violin plots is difficult to distinguish due to the grey dotted style. Enhancing its visibility would help readers interpret whether the distributions lie above or below the mean.

In summary, the study is methodologically sound, meets the expected standards of the field, and has substantial potential impact across several research domains, as well as practical implications for the development of more efficient transportation policies and models. The work can therefore be recommended for a publication after minor revision.

(Remarks on code availability)

The code package consists of four Python scripts accompanied by a folder containing all necessary data. It also includes a README file with installation instructions, dependency specifications, and a clear description of how each script corresponds to the methods presented in the paper. Overall, the provided code fully covers all methodological components of the study, and the complete dataset for the 30 evaluated cities is included. The code is well-structured, readable, and sufficiently documented. I was able to launch the code successfully after installing the necessary dependencies on a system with Windows 11 and Anaconda.

Version 1:

Reviewer comments:

Reviewer #1

(Remarks to the Author)

I would like to thank the authors for the substantial effort put into revising the manuscript. Nearly all of the concerns I raised in my previous review have been carefully addressed. I only ask the authors to consider two remaining minor points:

1) Figures 1 and 2 (readability). Regarding Figure 1, I wonder whether the geographic map is strictly necessary in the main text; it could be moved to the Supplementary Material without any loss of essential information. Even if the authors decide to keep the map in Figure 1, it seems redundant in Figure 2, as it is identical to that in Figure 1. Moreover, while the overall trends of the curves in Figure 2 are visible, the individual curves and numerical values are difficult to read because the plots are too small. Removing the map would free space to enlarge the city-specific plots and substantially improve their readability. In general, Figure 2 should be revised to make the panels larger and more legible.

2) Clarification of the Jaccard distance. It is still unclear how the Jaccard distance is computed in this work. The Jaccard measure is defined on two sets (as the ratio between their intersection and their union), but the manuscript does not clearly specify what the two sets are in this context. More generally, I believe that the procedure described in point 6 of the rebuttal should be explained in greater detail (either in the main text or in the Supplementary Material) to ensure full clarity and reproducibility.

(Remarks on code availability)

Reviewer #3

(Remarks to the Author)

The revised manuscript adequately addresses all points raised during review. The authors' responses are clear, and the changes made resolve the previous concerns. Specifically:

- the authors discuss the implications of relaxing the stationarity requirement in the proposed method, as well as the choice of parameters influencing the size of quasi-stationary neighborhoods;
- the authors discuss the influence of latent factors not included in the current feature set on the interpretation of the results;
- the proposed modifications improve the readability and completeness of the manuscript.

I therefore recommend acceptance of the manuscript for publication.

(Remarks on code availability)

Version 2:

Reviewer comments:

Reviewer #1

(Remarks to the Author)

The authors have thoroughly addressed all of my concerns. I appreciate the diligence and care taken in the revision.

(Remarks on code availability)

Response to Reviewers

Reviewer #1 (Remarks to the Author):

This paper introduces STCCM (spatio-temporal convergent cross mapping) to investigate bidirectional causal relationships between urban systems and traffic dynamics. The topic is timely and relevant: uncovering causal dependencies in urban processes remains a central yet underexplored challenge in urban science. The proposed approach has the potential to advance our understanding of how urban structure and traffic mutually influence each other. However, the presentation of the methodology and the results requires substantial improvement before the contribution can be properly assessed.

Response: We would like to express our sincere gratitude for your constructive assessment of our work, especially regarding the timeliness of the problem and the potential of STCCM. We also acknowledge your central concern that, in the original manuscript, key methodological ideas and parts of the computational pipeline were not presented intuitively enough in the main text, making it difficult for readers to assess the novelty and interpret the results.

In the revised manuscript, we substantially strengthened the presentation and interpretability of the manuscript. In detail, we (i) added a clear non-technical conceptual introduction of STCCM in the main text and clarified how it integrates with STWR; (ii) expanded and better positioned our work within the recent literature on urban systems and traffic dynamics; (iii) redesigned figures that were hard to interpret (most notably Fig. 1f and Fig. 2a) by separating city-level plots from the world map and improving legibility; (iv) added brief and intuitive definitions for non-obvious features at first mention; (v) made the statistical validation of cross-mapping skill (ρ) and directional asymmetry ($\Delta\rho$) explicit and easy to follow; and (vi) clarified previously under-specified technical elements (e.g., Jaccard similarity, DTW, embedding dimension, simplex projection) at their first appearance.

We believe these revisions have fundamentally improved the clarity and accessibility of the manuscript. We provide a point-by-point response to your detailed comments below.

Detailed comments

1) Several key concepts are fully delegated to the Methods section. While technical details can indeed be confined there, the main text should still provide intuitive explanations of the central ideas. This is particularly true for STCCM, which represents the core methodological innovation of the paper. Without a conceptual introduction in the main text, readers cannot grasp the novelty or rationale of the method.

Response: Thank you very much for this insightful comment! We fully agree that the main text should include intuitive explanations of the key methodological concepts so that readers can better understand their novelty and rationale. In the revised manuscript, we have made several changes to address this concern.

In the Introduction section, we have added an intuitive description of STCCM that conveys its

core idea in non-technical terms. This complements the formal methodological exposition and makes the underlying rationale accessible to readers unfamiliar with CCM-based approaches. We also clarify the respective roles of STWR and STCCM in the spatio-temporal causality framework, explaining how the two components jointly bridge correlation and causation. These changes can be found in Lines 63-72 on Page 3: *“Intuitively, STCCM uncovers causal relationships by examining whether the spatio-temporal evolution of one process, such as traffic dynamics during congestion periods, embeds information that allows the recovery of another process within the joint state space of interacting urban systems. The framework combines STCCM with spatio-temporal weighted regression (STWR) (see Methods). In detail, STWR first characterizes the spatially heterogeneous and temporally varying relationships between urban systems and traffic dynamics, and then synthesizes these locally estimated effects into composite indicators that capture the combined influence of urban structure, form, and function. Subsequently, STCCM uses these composite indicators to infer nonlinear and bidirectional causal pathways during congestion periods, when mobility-infrastructure mismatches amplify causal signals and make them more detectable and policy-relevant.”*

In the Result section for STWR, we revised the description of STWR to make the notions of spatial heterogeneity and temporal variation more transparent to readers in Lines 89-90 on Page 4: *“The STWR model is then employed to quantify these relationships, allowing the strength of association to vary across space and time rather than imposing a single global effect.”*

In the Result section for STCCM, before introducing Fig. 2, we now explicitly explain how to interpret the L - ρ plots and why the convergence of ρ with increasing library size L is regarded as evidence of causality, which provides the basis for understanding the strength and direction of causal effects. The revisions are in Lines 158-170 on Pages 7-8: *“Each L - ρ plot serves as a diagnostic tool for causal inference, illustrating how predictability evolves with increasing library size L . Here, L represents the number of observations used for state-space reconstruction, while ρ represents the cross-mapping skill of one variable based on another, i.e., how well one variable recovers the state of the other. For each city and each component of the urban system, we generate two L - ρ curves: one for urban systems \rightarrow traffic dynamics and one for traffic dynamics \rightarrow urban systems. Crucially, evidence of causality is established not by a static ρ value, but by the property of convergence: a curve that rises and stabilizes at higher ρ values as L grows. This trend indicates a robust causal signal, as additional observations improve coverage of the reconstructed state space and help reveal the underlying deterministic rules of the system [34]. Furthermore, the shaded regions between the two curves in Fig. 2(a) highlight the magnitude of asymmetry ($\Delta\rho$) between the bidirectional influences. Consequently, for a given urban system indicator X and traffic dynamics indicator Y , if the ρ values for $X \rightarrow Y$ consistently exceed those for $Y \rightarrow X$ as L increases, X is inferred to be the dominant causal driver of Y , and vice versa.”*

Taken together, these additions provide readers with intuitive and high-level explanations of STCCM and its integration with STWR directly in the main text, while leaving the mathematical

details in the Methods section. We believe these revisions address your concern and make the core methodological innovation of the paper much more transparent and interpretable.

2) The discussion would benefit from stronger engagement with recent studies on the interplay between urban form and traffic dynamics (e.g., <https://arxiv.org/abs/2510.02582>, <https://onlinelibrary.wiley.com/doi/10.1155/2023/6144048>, <https://www.sciencedirect.com/science/article/abs/pii/S0966692321002532>). These works provide useful benchmarks and could help position the proposed approach within the broader literature. The authors should systematically collect and discuss similar studies, emphasizing what STCCM adds conceptually or empirically.

Response: Thank you very much for this helpful suggestion! In the revised version, we have strengthened the manuscript's engagement with recent work on the interplay between urban systems and traffic dynamics. We expanded the literature coverage by incorporating the three benchmark studies you suggested and adding several closely related studies, and we revised the main text to summarize how these studies typically model urban systems and traffic dynamics. Building on this expanded context, we also clarified the conceptual and empirical contributions of STCCM, particularly its ability to provide bidirectional and asymmetric causal diagnostics that are comparable across cities and day types.

In the Introduction section, we have reframed the paragraph that reviews previous evidence linking built-environment factors to traffic dynamics, and we also briefly organize the commonly used modelling approaches. In addition, we explicitly incorporated the three benchmark studies (i.e., route diversification in road networks, POI-based road-function analysis, and street-network design effects), alongside other closely related studies, to provide a more systematic context. The revised content is shown in Lines 21-35 on Page 2: *"In particular, built-environment factors, ranging from urban morphology indices and land-use types to network structure, have been shown to exert significant influences on traffic states and congestion patterns across diverse metropolitan contexts [15, 18-20]. To investigate these complex interactions between urban systems and traffic dynamics, many methodological approaches have been adopted, including global statistical and econometric models for uncovering macro-level determinants and longitudinal dependencies [15, 19], local and spatio-temporal varying-coefficient models for characterizing spatiotemporal heterogeneity and dynamic associations [21, 22], and computational data-driven frameworks for capturing nonlinear dynamics and fine-grained mobility patterns [2, 3, 23]. Furthermore, such methodological advances have been deployed in detailed case studies to quantify route diversification capabilities within network topologies [24], assess the influence of road functions on localized congestion [25], and evaluate the impacts of street design and intersection density on traffic performance [18]. Collectively, these studies establish important empirical benchmarks on the interplay between urban systems and traffic dynamics, underscoring that both supply-side factors (e.g., network structure) and demand-side factors (e.g., land-use and activity patterns) within the built environment are*

critical determinants of traffic states.”

In the Discussion section, we have added a new paragraph to connect our findings with these benchmarks and to position STCCM within the broader literature. We highlight that most prior evidence remains association-based, whereas STCCM provides bidirectional and potentially asymmetric causal diagnostics and yields comparable causal fingerprints across cities and between rest and work days. Then, we also discuss the implications of these fingerprints for context-sensitive congestion mitigation strategies that coordinate land-use configuration, activity distribution, and network operations. The added paragraph is shown in Lines 334-345 on Page 14: *“Our empirical findings are broadly consistent with, yet also extend, previous evidence on the links between urban systems and traffic dynamics [18-20, 24, 25]. By moving from correlations to explicit causal inference, our spatio-temporal causality framework complements this literature in two ways. Conceptually, a large share of existing work relies on global or locally varying associations [15, 22], which are valuable for explanation and prediction but do not directly resolve directionality or feedback. In contrast, the STCCM model provides bidirectional and potentially asymmetric causal diagnostics, enabling a comparable assessment of how urban structure, form, and function exert directional influence on traffic dynamics across cities and between rest and work days. Empirically, the resulting causal fingerprints reveal recurring causal patterns and cross-city regularities, while also highlighting substantial context dependence. These findings suggest that interventions focusing solely on network supply may overlook critical demand-side mechanisms embedded in urban form and function. Consequently, effective mitigation requires a synergistic approach that coordinates land-use configuration and activity distribution with network operations, tailored to local causal pathways.”*

We believe these revisions provide a clearer and more up-to-date positioning of our study and strengthen the discussion of what STCCM adds beyond existing benchmark approaches.

References:

- [2] L. Pappalardo, E. Manley, V. Sekara, L. Alessandretti, Future directions in human mobility science, *Nature Computational Science* 3 (2023) 588–600. doi: 10.1038/s43588-023-00469-4.
- [3] J. Cao, W. Tu, R. Cao, Q. Gao, G. Chen, Q. Li, Untangling the association between urban mobility and urban elements, *Geo-spatial Information Science* 27 (2024) 1071–1089. doi: 10.1080/10095020.2022.2157761.
- [15] M. Wang, N. Debbage, Urban morphology and traffic congestion: Longitudinal evidence from us cities, *Computers, Environment and Urban Systems* 89 (2021) 101676. doi: 10.1016/j.compenvurbsys.2021.101676.
- [18] D.-a. Choi, R. Ewing, Effect of street network design on traffic congestion and traffic safety, *Journal of Transport Geography* 96 (2021) 103200. doi: 10.1016/j.jtrangeo.2021.103200.
- [19] M. M. Rahman, P. Najaf, M. G. Fields, J.-C. Thill, Traffic congestion and its urban scale factors: Empirical evidence from American urban areas, *International Journal of Sustainable Transportation* 16 (2022) 406–421. doi: 10.1080/15568318.2021.1885085.

- [20] D. Xiao, I. Kim, N. Zheng, Does built environment have impact on traffic congestion?—a bootstrap mediation analysis on a case study of Melbourne, *Transportation Research Part A: Policy and Practice* 190 (2024) 104297. doi: 10.1016/j.tra.2024.104297.
- [21] X. Ma, J. Zhang, C. Ding, Y. Wang, A geographically and temporally weighted regression model to explore the spatiotemporal influence of built environment on transit ridership, *Computers, Environment and Urban Systems* 70 (2018) 113–124. doi: 10.1016/j.compenvurbsys.2018.03.001.
- [22] H. Liu, W. Zhang, S. Wang, Z. Cheng, L. Wei, W. Huang, Exploring the effect of built environment on spatiotemporal evolution of traffic congestion using a novel GTWR model: a case study of Hefei, China, *Transportation Letters* 17 (2025) 869–880. doi: 10.1080/19427867.2024.2396773.
- [23] Z. Kan, D. Liu, X. Yang, J. Lee, Measuring exposure and contribution of different types of activity travels to traffic congestion using GPS trajectory data, *Journal of Transport Geography* 117 (2024) 103896. doi: 10.1016/j.jtrangeo.2024.103896.
- [24] G. Cornacchia, L. Pappalardo, M. Nanni, D. Pedreschi, M. C. González, A computational framework for quantifying route diversification in road networks, arXiv preprint arXiv:2510.02582 (2025). doi: 10.48550/arXiv.2510.02582.
- [25] H. Zhu, K. Zhang, C. Wang, L. Jia, S. Song, The impact of road functions on road congestions based on POI clustering: An empirical analysis in Xi'an, China, *Journal of Advanced Transportation* 2023 (2023) 6144048. doi: 10.1155/2023/6144048.

3) Some figures are difficult to interpret (e.g., Fig. 1f and Fig. 2a). Displaying all cities on a world map adds little to the understanding of the results. It would be clearer to present the graphs or scatter plots as standalone figures, perhaps selecting a subset of representative cities. This would considerably improve readability and allow the reader to focus on the patterns conveyed by the plots.

Response: Thank you for highlighting the readability issue. We agree that embedding many small plots within a global map made the original figures difficult to interpret, particularly for regions with dense cities (e.g., Europe and Southeast Asia). The combination of highly compressed panels and connecting leader lines also introduced visual clutter that obscured city-specific patterns.

To address this, we redesigned Fig. 1f and Fig. 2a by presenting the city-level results as standalone small-multiple panels arranged in a clean grid, rather than placing the plots on the world map. This redesign allows each panel to be substantially enlarged, making the STWR fingerprints in Fig. 1f and the L - ρ convergence curves in Fig. 2a clearly legible. To retain the worldwide context without sacrificing readability, we now include a simplified world map only as a spatial locator, showing the geographic distribution of the 30 cities without embedding any analytical plots. Cities are grouped by region using color-coded boxes, and the same colors are applied on the locator map, allowing immediate cross-referencing between the map and the corresponding panel groups. We also removed the crowded leader lines and labeled each city directly beneath its panel.

In addition, we have redrawn the individual L - ρ plots for each city in Fig. 2a, increasing the

font size and clarity of the x- and y-axis labels to ensure they are easily readable. Fig. 2a now also includes visual annotations (zoom-in indicators and $\Delta\rho$ shading cues) to clarify how the plotted elements should be interpreted. We also clarified this in the revised caption: “*The enlarged view on the right magnifies the trends at large library sizes to clearly reveal the directional difference in causal strength, while the shaded regions between the curves highlight the magnitude of asymmetry ($\Delta\rho$) between the bidirectional causal influences.*”

We considered showing only a subset of representative cities. However, this study discusses patterns across all 30 cities and is framed from a worldwide perspective. Since the new grid layout resolves the crowding problem while maintaining clarity, we retained the complete set of cities to provide a comprehensive view and to avoid any perception of selective presentation. This also allows readers to directly verify results for any city referenced in the main text without repeatedly consulting the Supplementary Information. The locator map remains intentionally minimal and is included solely for geographical context.

We believe these revisions directly address the concern about interpretability and enable readers to focus on the patterns conveyed by the plots. We implemented the same redesign principles for Extended Data Figs 1 and 2. The revised figures of Fig. 1f and Fig. 2a are as follows.

4) The meaning of some features is not immediately intuitive. For instance, SPRT (sports area) is self-explanatory, but CONTAG (contagion) is not. The main text should briefly describe each feature the first time it is mentioned, so that the discussion can be followed without repeatedly consulting the Supplementary Information.

Response: We appreciate your insightful suggestion to provide brief descriptions for features in the main text. Following this, we have revised the main text and added intuitive explanations for features that are not self-explanatory when they first appear, including CRS_CT, RD_DENS, NP, CONTAG, TE, ED, and SHEI. These revisions are primarily located in the Results section (Pages 5-7). For clarity, the brief explanations are summarized below:

- CRS_CT: *crossing count, the number of road intersections* (Line 131, Page 5).
- RD_DENS: *road density, the total road length per unit area* (Line 131, Page 5).
- NP: *number of patches, the total number of land-use patches* (Line 133, Page 5).
- CONTAG: *contagion, the degree to which land-use patches are clumped or dispersed* (Line 134, Page 5).
- TE: *total edge, the total length of all patch edges* (Line 137, Page 7).
- ED: *edge density, the total length of all patch edges per unit area* (Line 137, Page 7).
- SHEI: *Shannon's evenness index, the evenness of area distribution among patch types* (Fig. 1, Page 6).

To further support readability, we have also added Supplementary Table 9, which provides concise descriptions for all features used to characterize urban structure, form, and function.

Supplementary Table 9. *Descriptions of features characterizing urban structure, form, and function*

Abbreviation	Full Name	Definition
NODE_CT	Node Count	The number of nodes in the road network.
NODE_DENS	Node Density	The number of nodes per unit area.
CRS_CT	Crossing Count	The number of road intersections.
K_AVG	Average Node Degree	The average degree of nodes in the road network.
EDGE_LEN	Total Length	The total length of road segments.
RD_DENS	Road Density	The total road length per unit area.
NET_INT	Network Intensity	The total length of road segments normalized by the cell perimeter.
BTW_CEN	Average Betweenness Centrality	The average of the fraction of shortest paths that pass through each node.
CLS_CEN	Average Closeness Centrality	The average reciprocal of the sum of the shortest path distances from a node to all other nodes.
DEG_CEN	Average Degree Centrality	The average normalized degree of nodes within the cell.
TA	Total Area	The total area of land-use patches.
NP	Number of Patches	The total number of land-use patches.
PD	Patch Density	The number of land-use patches per unit area.
TE	Total Edge	The total length of all patch edges.
ED	Edge Density	The total length of all patch edges per unit area.

LSI	Landscape Shape Index	The measure of landscape shape complexity.
PARA	Perimeter-Area Ratio	The average perimeter-area ratio across all patches within the cell.
SHAPE	Shape Index	The average shape index across all patches, quantifying how much patch shapes deviate from compact forms.
FRAC	Fractal Dimension	The average fractal dimension of all patches, describing how perimeter complexity scales with patch size.
CONTAG	Contagion	The degree to which land-use patches are clumped or dispersed.
AI	Aggregation Index	The measure indicating how closely land-use patches are clustered together.
COHES	Cohesion	The physical connectedness of land-use patches within the cell.
SPLIT	Splitting Index	The fragmentation reflecting the degree of landscape subdivision.
PR	Patch Richness	The total number of distinct patch types.
SHDI	Shannon's Diversity Index	The diversity of area distribution among patch types.
SHEI	Shannon's Evenness Index	The evenness of area distribution among patch types.
TRNS	Transportation Area	The percentage of land designated for transport infrastructure.
COMM	Commercial and Business Facilities Area	The percentage of land used for commerce, finance, hospitality, and retail.
IND	Industrial Area	The percentage of land used for industrial production and extraction.
RES	Residential Area	The percentage of land designated for housing and living quarters.
FRST	Farmland and Forest Area	The percentage of natural, semi-natural, and productive rural land.
EDU	Education and Science Area	The percentage of land dedicated to educational and research institutions.
SPRT	Sports Area	The percentage of land used for physical sports and athletic activities.
PARK	Park Area	The percentage of public green spaces and manicured vegetation.
RECR	Recreation Area	The percentage of land used for leisure, tourism, and entertainment.
WATR	Water Area	The percentage of land covered by water bodies.
HLTH	Health Care Area	The percentage of land used for medical and care services.
CULT	Cultural Facilities Area	The percentage of land used for cultural arts and public gatherings.

These additions ensure that key features are understandable directly within the main text, while the Supplementary Information offers a reference for readers who wish to explore all features in detail.

5) Significance of ρ values in L- ρ analysis (Fig. 2b-d). In Figs. 2b-d, most cities appear close to the diagonal, suggesting that the estimated ρ values may not be statistically significant. The authors should clarify whether a threshold for the significance of ρ exists, how it is computed, and whether the reported relationships pass such a test. This is an important aspect for assessing the robustness of the findings.

Response: Thank you for this important observation! We note that proximity to the 45° diagonal in Fig. 2b-d indicates that the two directional cross-mapping skills are similar in magnitude (i.e., weak asymmetry), rather than implying that the ρ values themselves are close to zero or statistically non-significant. To explicitly address statistical significance and robustness, we conducted formal tests for (i) the significance of ρ and (ii) the significance of directional asymmetry ($\Delta\rho$). All tests are two-sided with a significance threshold of 0.05.

First, regarding the significance of ρ , our statistical validation was originally detailed only in the Supplementary Information and not sufficiently highlighted in the main text. We appreciate you pointing this out, as it is indeed crucial for assessing robustness. To clarify whether a threshold for the significance of ρ exists, we conducted a two-sided one-sample t-test on the distribution of ρ values across all BVCells. We find that the estimated ρ values for all 30 cities (across urban structure, form, and function) are statistically significant ($p < 0.001$; see Supplementary Tables 5 and 6).

Second, to address the concern that points near the diagonal might imply indistinguishable bidirectional effects, we explicitly tested the significance of the asymmetry using the difference in their ρ values, i.e., $\Delta\rho = \rho_{\text{urban systems} \rightarrow \text{traffic dynamics}} - \rho_{\text{traffic dynamics} \rightarrow \text{urban systems}}$. We performed two-sided paired Student's t-tests to compare BVCell-wise ρ values in both directions for each city. As detailed in the newly added Supplementary Table 7, the asymmetry is statistically significant for the majority of cities, even those visually close to the diagonal. For example, on rest days, significant asymmetry was observed in 26, 29, and 27 out of 30 cities for urban structure, form, and function, respectively. Notably, among these significant cases, the majority surpass the high significance threshold of $p < 0.001$.

The revisions in the manuscript are summarized as follows:

In the Methods section, we explicitly described the one-sample and paired Student's t-tests in Lines 656-664 on Pages 26-27: *“In addition, we performed statistical significance tests on the cross-mapping skill derived from the STCCM model. As our analysis covers the full study area partitioned into BVCells, we utilize the distribution across these spatial units as a standardized basis for statistical testing. First, the statistical significance of ρ at each library size was evaluated using a one-sample Student's t-test across all BVCells (see Supplementary Tables 5 and 6). Second, to verify*

the observed asymmetry, we assessed the significance of the difference between directional causal strengths ($\Delta\rho = \rho_{X \rightarrow Y} - \rho_{Y \rightarrow X}$) at the largest library size. This was conducted using a paired Student's *t*-test comparing the distributions of causal strengths in both directions across all BVCells for each city (see Supplementary Table 7). All tests reported are two-sided with a significance threshold of 0.05.”

In the Results section, we discussed the statistical significance of both ρ and $\Delta\rho$ to support the interpretation of Fig. 2 in Lines 201-208 on Pages 9-10: “Although some cities near the diagonal might suggest a balanced interaction, statistical tests support the robustness of these relationships. Specifically, the estimated causal strengths (ρ) are statistically significant across all 30 cities ($p < 0.001$; see Supplementary Table 5). Furthermore, despite the visual proximity to the diagonal, paired *t*-tests reveal that the asymmetry ($\Delta\rho$) remains statistically significant for the majority of cities on rest days, specifically in 26, 29, and 27 out of 30 cities for urban structure, form, and function, respectively (see Supplementary Table 7). Together, these results statistically support the conclusion that urban systems more often exert a dominant influence on traffic dynamics, consistent with the asymmetric nature of the causality.”

We also added Supplementary Table 7 reporting the $\Delta\rho$ values and their significance.

Supplementary Table 7. Statistical assessment of the causal asymmetry ($\Delta\rho$) between urban systems and traffic dynamics across 30 cities. The table reports the mean difference in cross-mapping skill at the largest library size, defined as $\Delta\rho = \rho_{\text{urban systems} \rightarrow \text{traffic dynamics}} - \rho_{\text{traffic dynamics} \rightarrow \text{urban systems}}$. A positive value indicates that urban systems exert a stronger causal influence on traffic dynamics, while a negative value implies the reverse. Statistical significance was evaluated using a paired Student's *t*-test comparing the distributions of causal strengths in both directions across all BVCells for each city, with two-sided *p*-values reported.”

City	Rest days			Work days		
	Structure $\Delta\rho$	Form $\Delta\rho$	Function $\Delta\rho$	Structure $\Delta\rho$	Form $\Delta\rho$	Function $\Delta\rho$
London	0.011***	0.071***	0.054***	-0.002	0.052***	0.013***
Paris	0.013***	0.046***	0.076***	0.016***	0.048***	0.063***
Zurich	-0.011	0.047***	0.059***	-0.045***	-0.096***	-0.030***
Rome	0.052***	0.077***	0.090***	0.084***	0.031***	0.110***
Vienna	0.004	-0.004	0.014***	0.011**	0.040***	0.024***
Berlin	0.010***	0.028***	0.042***	-0.019***	0.063***	0.021***
Stockholm	0.039***	0.049***	0.033***	0.004	0.028***	0.030***
Helsinki	0.019***	0.031***	0.018***	0.016***	0.005*	0.018***
St. Petersburg	0.048***	0.079***	0.086***	0.054***	0.109***	0.079***
Moscow	0.033***	0.109***	0.065***	0.030***	0.076***	0.058***
Riyadh	0.063***	0.132***	0.101***	0.062***	0.119***	0.134***
Dubai	0.019***	0.044***	0.043***	0.021***	0.043***	0.015***
New Delhi	0.054***	0.122***	0.120***	0.073***	0.165***	0.133***
Mumbai	0.015***	-0.023***	0.097***	0.005*	-0.053***	0.072***
Bangkok	0.055***	0.024***	0.067***	0.043***	0.033***	0.068***
Hong Kong	-0.010***	0.018***	0.000	-0.061***	-0.011***	-0.023***

Manila	0.030***	-0.072***	0.036***	-0.048***	0.021***	-0.068***
Kuala Lumpur	0.063***	0.055***	0.132***	0.093***	0.105***	0.123***
Singapore	0.043***	0.090***	0.063***	0.071***	0.122***	0.087***
Jakarta	0.056***	0.030***	-0.001	0.054***	0.046***	-0.003ns
Sydney	0.157***	0.139***	0.186***	-0.037***	-0.059***	-0.054***
Christchurch	-0.054***	0.020***	-0.005	-0.081***	-0.037***	0.028***
Los Angeles	0.005***	0.042***	-0.014***	-0.021***	0.026***	-0.019***
Chicago	0.023***	0.078***	0.087***	0.020***	0.040***	0.031***
Toronto	0.015***	0.041***	0.025***	-0.015***	0.016***	0.030***
Mexico City	-0.005**	0.009***	0.023***	0.060***	0.056***	0.225***
Lima	0.001	-0.004*	0.015***	0.019***	-0.008***	0.029***
Buenos Aires	0.049***	0.093***	0.116***	-0.007**	0.034***	0.036***
Rio de Janeiro	-0.002	-0.017***	0.044***	-0.006**	0.027***	-0.035***
Cape Town	-0.013***	0.069***	0.055***	0.019***	0.023***	-0.025***

Finally, we updated the captions for Fig. 2 and Supplementary Tables 5 and 6 to reflect the statistical method. In Fig. 2, we added the statement: “*Note that all plotted cities exhibit statistically significant causal relationships (ρ) at the $p < 0.001$ level. Detailed significance tests for both ρ and the directional asymmetry ($\Delta\rho$) are provided in Supplementary Tables 5 and 7.*” In Supplementary Tables 5 and 6, we specified: “*Statistical significance was evaluated using a one-sample Student's t-test on the distribution of ρ values across all BVCells at each library size, with two-sided p-values reported.*”

These tests clarify that although cities appear close to the diagonal, the estimated ρ values are statistically significant. Furthermore, these results validate the existence of the bidirectional yet asymmetric causality between urban systems and traffic dynamics reported in our study.

6) At line 202, the paper mentions the “Jaccard similarity” but does not specify what entities are being compared. This issue occurs in several instances: key technical aspects are mentioned without sufficient explanation, making the paper difficult to follow for readers unfamiliar with the specific computational pipeline.

Response: Thank you very much for pointing out this critical issue. We have revised the main text to clarify how the Jaccard-based measure is used in our framework. Specifically, we now state that the Jaccard distance measures the dissimilarity between the enclosed areas under the bidirectional L - ρ causality curves of two cities, thereby reflecting differences in the overall magnitude of their causal influence. We also carefully reviewed the manuscript to identify similar cases where key technical aspects were mentioned without sufficient explanation. Following your suggestion, we now provide brief descriptions at the first appearance of the following concepts in the main text:

- Jaccard similarity: “*In contrast, the Jaccard distance quantifies the dissimilarity in the enclosed areas of these bidirectional L - ρ curves, reflecting differences in the overall magnitude of their causal influence.*” (Lines 238-240, Pages 10-11).

- Dynamic time warping: *“Specifically, dynamic time warping (DTW) evaluates the shape difference between the bidirectional L- ρ curves of two cities, capturing variations in the temporal patterns of causality.”* (Lines 236-238, Page 10).
- Embedding dimension: *“Thus, the embedding dimension E determines how many spatial-lagged neighbors are included in the reconstructed state space, where a larger E corresponds to higher-order spatial neighborhoods. The sensitivity analysis of STCCM with varying E is reported in Supplementary Figs. 2 and 3.”* (Lines 614-617, Pages 24-25).
- Simplex projection: *“Simplex projection estimates the target variable by locating its nearest reconstructed states and forming a distance-weighted average of their observed values.”* (Lines 624-626, Page 25).

Together with the revisions made in response to Comments 1 and 4, we believe that the revised manuscript now allows readers who are less familiar with the computational pipeline to follow the methodological logic directly from the main text.

In summary, the paper tackles an important and underexplored topic with a potentially valuable methodological contribution, but the clarity, contextualization, and interpretability of both the methods and results must be significantly improved.

Response: Thank you very much for this concluding remark. We appreciate your recognition of the study’s potential value, and we have focused our revision on the three areas you highlighted: clarity (via intuitive conceptual explanations), contextualization (via benchmarking with recent literature), and interpretability (via figure redesign and rigorous statistical validation). We believe these improvements now allow readers to assess methodological contribution and the results more clearly.

Reviewer #1 (Remarks on code availability):

The repository's README file describes the code structure, data organization, and system requirements necessary to replicate the study. Although I did not attempt a full replication, the repository appears to provide sufficient information and resources to enable reproducibility.

Response: Thank you for your remarks on code availability and reproducibility. We appreciate your careful check of the repository documentation and your assessment that the README provides sufficient details to support replication.

Reviewer #3 (Remarks to the Author):

The current landscape of urban simulations, predictive, and optimization models is largely fragmented, with a predominant presence of solutions tailored to a specific cities or regions. Significant differences in urban layouts and traffic dynamics hamper the easy transfer of models between the cities. In modern conditions, when the pace of change in city infrastructure and mobility patterns can be compared to the pace of model development itself, the ability to accelerate the

process by exploiting and fine-tuning results from similar cities becomes crucial. How can such similarity be defined and empirically evaluated so that cities can be meaningfully partitioned into clusters with similar traffic dynamics, and, moreover, similar types of interventions to reduce congestion?

The work answers this question for the first time by considering not only a single aspect (e.g. topology of the road network) but the interplay between structure, form, function and dynamics. This makes it highly significant not only for urban studies and geospatial science, but also for Urban AI applications, suggesting a set of interpretable variables that can serve as a context which conditions decision-making policies and facilitates transfer learning between cities. The latter is also supported by the city causal archetypes revealed in the analysis, which enable the identification of cities with similar coupling between urban systems and traffic dynamics. A second aspect defining the significance of the study is its focus on causation rather than correlation. This provides a foundation for scientifically justified transportation policies and interventions, as well as for designing assumptions and induction biases for intelligent transportation models which both are highly relevant and challenging problems in the field of intelligent transportation systems. In summary, this work is highly significant for several research fields including urban studies, geospatial science, intelligent transportation systems and data-driven traffic simulation.

Response: Thank you very much for this thoughtful and encouraging assessment! We really appreciate your recognition that advancing transferable urban modelling requires defining similarity across cities in a way that moves beyond single aspects of the built environment. In line with your remarks, our work integrates urban structure, form, and function with traffic dynamics to derive cross-city causal archetypes.

We also appreciate your emphasis on the importance of moving from correlation to causation. By providing bidirectional and potentially asymmetric causal diagnostics that are comparable across cities and day types, we aim to offer a more robust foundation for scientifically justified congestion mitigation and for the development of transferable Urban AI and intelligent transportation models.

The work is grounded in well-established literature on causal inference, spatio-temporal prediction models, geospatial information science and urban mobility studies. The application of Convergent Cross Mapping (CCM) to detect (potentially bidirectional) causalities is methodologically sound. The original CCM method assumes that the underlying dynamical system is stationary, so that the functional relationships between observables remain constant over time. However, this assumption may not hold for the systems under study which are characterized by spatial heterogeneity and temporal variability. The authors relax the strict stationarity assumption (similarly to GCCM but extending to both spatial and temporal dimensions) and propose novel ST-CCM method for bidirectional causal estimation in spatio-temporal systems. ST-CCM combines spatio-temporal CCM with spatio-temporal weighted regression (STWR) that smooths the original data into quasi-stationary, piecewise regions. The approach appears both methodologically innovative and

practically sound, as it denoises the data and transforms them into a set of interpretable composite indicators which tend to be more stationary across space-time cells than the original variables. Since the data points to calculate the estimates of ρ are derived from locally fitted STWR regression models, a brief discussion on the theoretical and practical implications of relaxing the stationarity assumption (e.g. how the bandwidth of the STWR kernel or buffer size of Voronoi cells influence the convergence of ST-CCM, or how to interpret the results for the cities comprising sub-regions with highly heterogeneous structure, form or function), would further strengthen the methodological contribution of the study.

Response: Thank you very much for this insightful comment! We agree that a discussion on the theoretical and practical implications of relaxing the stationarity assumption would significantly strengthen the manuscript. In the revised manuscript, we added a concise discussion to clarify how our framework relaxes strict stationarity through STWR-based smoothing and what this implies for the interpretation of STCCM results.

First, after constructing the STWR-based composite indicators, we now explicitly describe the resulting quasi-stationary approximation and clarify that STCCM should be interpreted as a BVCell-level causal diagnostic defined within the local support of the spatio-temporal kernel and the BVCell aggregation (Lines 572-584 on Page 23): *“In urban systems, strict stationarity is rarely plausible due to the inherent spatial heterogeneity and temporal non-stationarity of interactions between the built environment and traffic dynamics [21, 22, 66]. Consequently, our framework relaxes the stationarity requirement by first estimating spatially and temporally varying coefficients with STWR and then smoothing these coefficients with a bi-square kernel to construct composite indicators $CI(s,t)$ [67, 69]. This procedure yields quasi-stationary neighborhoods in which the association structure is approximately stable within the local support defined by the STWR kernel and the BVCell aggregation. Practically, the STWR bandwidth controls the bias-variance trade-off: narrower bandwidths preserve fine-grained heterogeneity but may retain more local noise, while wider bandwidths improve stability but may blend distinct local regimes. In addition, the BVCell buffer size determines the spatial support over which traffic and surrounding urban contexts are aggregated, thereby affecting the homogeneity of each cell. The choice of a 500-meter buffer is consistent with prior evidence on the scale of local traffic impacts [5, 16]. Increasing this size would likely increase signal stability through aggregation but decrease the sensitivity to micro-scale urban system variations.”*

Second, after introducing the L - ρ convergence criterion and the library-size design, we added a short paragraph discussing how STWR bandwidths and BVCell buffer sizes can affect the stability and convergence of L - ρ curves by trading off local heterogeneity against denoising and regime mixing (Lines 645-653, Page 26): *“Since STCCM inputs are derived from locally fitted and smoothed STWR models, key scale parameters can influence the convergence behavior of the cross-mapping skill ρ with increasing library size L . Intuitively, a narrower STWR bandwidth (or finer BVCell partition) yields more localized and potentially more heterogeneous composite-indicator*

fields [67, 69], which can reduce the effective signal-to-noise ratio and lead to slower or less stable convergence in L - ρ curves. Conversely, overly large bandwidths (or coarse spatial aggregation) may blur distinct sub-regimes and attenuate causality signals by mixing mechanisms, producing flatter L - ρ curves and weaker separations between $\rho_{X \rightarrow Y}$ and $\rho_{Y \rightarrow X}$. Therefore, the selected bandwidth and BVCell buffer size should be interpreted as defining the operative scale of quasi-stationarity, within which the local causal diagnostics are valid.”

Third, in the Discussion section, we clarified how to interpret city-level causal fingerprints for cities comprising highly heterogeneous sub-regions, emphasizing that the reported patterns summarise mixtures of BVCell-level regimes and motivating stratified extensions for sub-regional diagnosis (Lines 325-334, Page 14): *“In this context, it is important to note that our STCCM causal fingerprints aggregate local causal diagnostics, computed over within-city spatial units, into city-level L - ρ patterns. For cities with strongly heterogeneous sub-regions, such as pronounced core-periphery contrasts or polycentric structures, the city-level curves may reflect a mixture of distinct local regimes rather than a single dominant mechanism. In these instances, weaker separations between bidirectional cross-mapping skills or less pronounced convergence should be interpreted cautiously, as they may arise from offsetting local causal pathways. This motivates future stratified extensions in which STCCM is applied separately across spatial-unit groups defined by congestion intensity, centrality, or urban form archetypes. Such an approach would better reveal sub-regional causal heterogeneity while maintaining cross-city comparability through a consistent spatial support and kernel design.”*

We believe these additions strengthen the methodological contribution by making the quasi-stationarity assumption, its scale dependence, and its interpretive implications explicit in the main text.

In the study, STWR models are estimated for three groups of variables representing urban structure, form, and function, and a single response variable related to traffic congestion. This enables a comparative evaluation of how well congestion can be explained by the composite indicators across cities worldwide. The results demonstrate that the proposed indicators have strong and significant correlation with traffic dynamics, with urban structure contributing more than form or function. A short discussion of the variable selection for composite indicators would further strengthen this result. For example, if the chosen variables primarily reflect certain types of cities, the regression model may exhibit lower explanatory power not only because the underlying processes differ, but also because the feature set is less representative for other urban contexts.

Response: We sincerely appreciate your valuable suggestion! We agree that cross-city differences in STWR explanatory power can be influenced not only by heterogeneous underlying processes, but also by how representative the selected feature sets are for different urban contexts. To address this, we have strengthened the manuscript in three places.

First, in the Methods section, we added a concise rationale for the variable selection used to

generating the composite indicators for urban systems, emphasizing global comparability, interpretability, and consistent computability from OSM data, while also clarifying the potential representativeness boundary of a fixed feature library: *“To ensure cross-city comparability, we compute these urban system features from globally available OSM data and select variables based on three criteria: (i) representing complementary aspects of the three components; (ii) established interpretability in existing urban and transport studies; and (iii) consistent computability at the BVCell scale across cities.”* (Lines 481-484, Page 20) and *“However, since these composite indicators are derived from a globally consistent feature set, cross-city differences in STWR model fit may reflect not only heterogeneous urban-traffic interactions but also the degree to which the selected variables capture locally salient determinants of traffic patterns.”* (Lines 569-571, Page 23).

Second, in the Results section, we have refined our interpretation of cross-city differences in R^2 . We now explicitly acknowledge that lower R^2 values may arise not only from weaker couplings between form/function and traffic dynamics, but also from the extent to which the globally consistent feature set captures locally salient determinants of traffic patterns. The revised text states *“These disparities appear to be partly shaped by underlying socioeconomic conditions and governance context [38], and also by the extent to which the globally consistent feature set captures locally salient determinants of traffic patterns.”* (Lines 108-109, Page 5) and *“In these contexts, the lower R^2 indicates that the form and function features used here account for a smaller share of the variance in traffic patterns. This may reflect weaker associations with spatial configurations and land-use distributions, or the influence of latent factors not captured by the current feature set used to construct the composite indicator.”* (Lines 117-121, Page 5).

Third, in the Discussion, we explicitly stated this representativeness issue as a limitation and outlined future directions to expand the feature library and consider more context-sensitive specifications (Lines 354-360, Page 15): *“Second, the composite indicators for each urban system component are constructed from a fixed set of globally available features to ensure cross-city comparability. While this design facilitates consistent benchmarking, it may underrepresent context-specific determinants of congestion in some cities (e.g., multimodal supply, demand-management policies, or informal transport), which can lower the explanatory power of the association models even when strong urban-traffic coupling exists. Future work could expand the feature set and explore hierarchical specifications or models tailored to specific city clusters to obtain more representative composite indicators across diverse urban contexts.”*

These additions strengthen the interpretation of the composite indicators and clarify the sources of cross-city variation in model fit. We believe the revised discussion better situates our framework within diverse global urban contexts and addresses your concern.

The third important result of the study is the development of hierarchical clustering approach that groups the cities according to the direction and strength of causality between urban structure, form,

and function, and traffic dynamics), using ST-CCM outputs (L - ρ curves) as input data. The clustering results reveal the existence of three distinct causal archetypes of cities. This finding is highly relevant for scientifically grounded design of transportation policies and interventions, as the identified direction and strength of causality in a city highlights specific groups of measures that could be the most effective in mitigating congestion.

The work supports all its conclusions and claims, and no additional evidence is needed. In particular, the goodness of fit of the STWR regression for modelling the traffic indicator is supported by consistently high R^2 and AICc values compared to the GWR model. The authors evaluate the significance of ρ values for two library sizes, which aligns with standard CCM validation practice (see, e.g., Gao, B., Yang, J., Chen, Z. et al. Causal inference from cross-sectional earth system data with geographical convergent cross mapping. *Nat Commun* 14, 5875 (2023)). The general conclusions regarding the causal interplay between urban systems and traffic dynamics, as well as the explanatory power of the urban structure, form, and function composite indicators, are well supported by the extensive dataset covering 30 cities on multiple continents. Moreover, the uniformity of the traffic congestion data which are sourced from the same provider and tested for spatial and temporal representativeness, adds credibility to the reported findings.

Response: We sincerely thank you for your positive assessment! We are really encouraged by your recognition that these findings are highly relevant for the scientifically grounded design of transportation policies and interventions. We also appreciate your note that the main claims are well supported by the STWR & GWR performance comparisons, standard CCM significance validation practice (e.g., Gao et al., 2023), and the credibility of the cross-city findings.

The study provides enough details in the methods to ensure reproducibility. As a minor suggestion, in Figure 1c-e, the zero reference line in the violin plots is difficult to distinguish due to the grey dotted style. Enhancing its visibility would help readers interpret whether the distributions lie above or below the mean.

Response: We appreciate this valuable feedback regarding visual clarity. In the revised Figure 1c-e, we have enhanced the visibility of the zero reference line by changing it to a bold black line. In addition, to prevent any visual confusion between this reference line and the internal data markers, we have updated the inner quartile lines within the violin plots to white. These adjustments ensure high contrast against the background, allowing readers to effortlessly distinguish whether the coefficient distributions lie above or below the mean.

The revised figure is shown as follows:

In summary, the study is methodologically sound, meets the expected standards of the field, and has substantial potential impact across several research domains, as well as practical implications for the development of more efficient transportation policies and models. The work can therefore be recommended for a publication after minor revision.

Response: Thank you very much for this positive overall assessment! Your endorsement reinforces our confidence in the validity and impact of this work. We have carefully addressed all the minor revisions suggested to further strengthen the manuscript.

Reviewer #3 (Remarks on code availability):

The code package consists of four Python scripts accompanied by a folder containing all necessary data. It also includes a README file with installation instructions, dependency specifications, and a clear description of how each script corresponds to the methods presented in the paper. Overall, the provided code fully covers all methodological components of the study, and the complete dataset for the 30 evaluated cities is included. The code is well-structured, readable, and sufficiently documented. I was able to launch the code successfully after installing the necessary dependencies on a system with Windows 11 and Anaconda.

Response: Thank you very much for your detailed remarks on code availability. We appreciate your assessment that the package is well structured and sufficiently documented, and we are grateful that you were able to install the dependencies and successfully launch the workflow on your system, which further supports the reproducibility of our study.

Response to Reviewers

Reviewer #1 (Remarks to the Author):

I would like to thank the authors for the substantial effort put into revising the manuscript. Nearly all of the concerns I raised in my previous review have been carefully addressed. I only ask the authors to consider two remaining minor points:

1) Figures 1 and 2 (readability). Regarding Figure 1, I wonder whether the geographic map is strictly necessary in the main text; it could be moved to the Supplementary Material without any loss of essential information. Even if the authors decide to keep the map in Figure 1, it seems redundant in Figure 2, as it is identical to that in Figure 1. Moreover, while the overall trends of the curves in Figure 2 are visible, the individual curves and numerical values are difficult to read because the plots are too small. Removing the map would free space to enlarge the city-specific plots and substantially improve their readability. In general, Figure 2 should be revised to make the panels larger and more legible.

Response: Thank you for this helpful follow-up. We agree that repeating the geographic map in Fig. 2 was redundant and that it constrained the readability of the city-level L - ρ panels. We therefore revised the figure design to enhance legibility.

In the revised manuscript, we removed the world map from Fig. 2 and reorganized the L - ρ curves into a larger grid (with fewer panels per row), which substantially increases the panel size and allows a larger font for axis and tick labels. To further improve readability, we increased the line weights of the individual curves and added subtle gridlines to facilitate reading numerical values. The geographic locator map is now shown only in Fig. 1 to avoid duplication.

These changes make both the individual curves and their numerical scales clearly interpretable. We applied the same redesign to Extended Data Fig. 2. The revised Fig. 2a is provided below.

2) Clarification of the Jaccard distance. It is still unclear how the Jaccard distance is computed in this work. The Jaccard measure is defined on two sets (as the ratio between their intersection and their union), but the manuscript does not clearly specify what the two sets are in this context. More generally, I believe that the procedure described in point 6 of the rebuttal should be explained in greater detail (either in the main text or in the Supplementary Material) to ensure full clarity and reproducibility.

Response: Thank you for this suggestion. We revisited our description and agreed that it did not explicitly define the two sets used in the Jaccard computation. In the revised manuscript, we now clarify that the Jaccard distance is computed on the 2D regions under the bidirectional L - ρ curves, where the bidirectional region for each city is constructed as the union of the two directional regions ($X \rightarrow Y$ and $Y \rightarrow X$). We then compute the Jaccard index as the intersection-over-union (IoU) of these regions and define the Jaccard distance as $1 - J(\cdot)$.

Accordingly, we added a concise clarification in the Results stating that the Jaccard distance is based on the IoU of the bidirectional regions (Lines 238-240, Page 11): “*In contrast, the Jaccard distance measures dissimilarity via the intersection-over-union (IoU) of the regions under the bidirectional L - ρ curves (defined as the union of the two directional regions), reflecting differences in the overall magnitude of their causal influence.*”

We also added a detailed definition in the Methods that explicitly specifies the two sets, their union construction, and the IoU formula, together with a description of the practical implementation (Lines 679-689, Pages 27-28): “*Similarly, we quantify magnitude differences using a Jaccard distance defined on regions under the bidirectional L - ρ curves over $L \in [L_{min}, L_{max}]$, where $[L_{min}, L_{max}]$ denotes the library-size range considered in STCCM. For each city m and dimension $X \in \{X_{str}, X_{frm}, X_{fun}\}$ on rest days r , we define each directional region as the area between the curve and the L -axis (i.e., $u = 0$):*

$$A_{X \rightarrow Y}^{(m,r)} = \{(L, u): L \in [L_{min}, L_{max}], 0 \leq u \leq P_{X \rightarrow Y}^{(m,r)}(L)\},$$

$$A_{Y \rightarrow X}^{(m,r)} = \{(L, u): L \in [L_{min}, L_{max}], 0 \leq u \leq P_{Y \rightarrow X}^{(m,r)}(L)\}.$$

In practice, each directional region is represented as a polygon by connecting sampled points $(L_i, \rho(L_i))$ and closing the boundary with $u = 0$. The bidirectional region is then defined as

$$A_X^{(m,r)} = A_{X \rightarrow Y}^{(m,r)} \cup A_{Y \rightarrow X}^{(m,r)}.$$

Given two cities m and n , the Jaccard index $J(\cdot)$ is computed as the intersection-over-union of these bidirectional regions:

$$J(A_X^{(m,r)}, A_X^{(n,r)}) = \frac{|A_X^{(m,r)} \cap A_X^{(n,r)}|}{|A_X^{(m,r)} \cup A_X^{(n,r)}|},$$

and the Jaccard distance is $JaccardDist = 1 - J(\cdot)$. Accordingly, the overall Jaccard distance between cities m and n on rest days r is computed as follows:

$$Jaccard_{(m,n,r)} = \sum_{X \in \{X_{str}, X_{frm}, X_{fun}\}} JaccardDist(A_X^{(m,r)}, A_X^{(n,r)}).”$$

We believe these revisions address the concern by explicitly defining the two sets used in the Jaccard computation and ensuring clarity and reproducibility.

Reviewer #3 (Remarks to the Author):

The revised manuscript adequately addresses all points raised during review. The authors' responses are clear, and the changes made resolve the previous concerns. Specifically:

- the authors discuss the implications of relaxing the stationarity requirement in the proposed method, as well as the choice of parameters influencing the size of quasi-stationary neighborhoods;
- the authors discuss the influence of latent factors not included in the current feature set on the interpretation of the results;
- the proposed modifications improve the readability and completeness of the manuscript.

I therefore recommend acceptance of the manuscript for publication.

Response: We sincerely thank you for your supportive evaluation and recommendation for acceptance. We are very pleased that the revisions have addressed the concerns and enhanced the manuscript's clarity and completeness.